# FedGEMS: Federated Learning of Larger Server Models via Selective Knowledge Fusion

## Abstract

Today data is often scattered among billions of resource-constrained edge devices with security and privacy constraints. Federated Learning (FL) has emerged as a viable solution to learn a global model while keeping data private, but the model complexity of FL is impeded by the computation resources of edge nodes. In this work, we investigate a novel paradigm to take advantage of a powerful server model to break through model capacity in FL. By selectively learning from multiple teacher clients and itself, a server model develops in-depth knowledge and transfers its knowledge back to clients in return to boost their respective performance. Our proposed framework achieves superior performance on both server and client models and provides several advantages in a unified framework, including flexibility for heterogeneous client architectures, robustness to poisoning attacks, and communication efficiency between clients and server on various image classification tasks.

## 1 Introduction

Nowadays, powerful models with tremendous parameters trained with sufficient computation power are indispensable in Artificial Intelligence (AI), such as AlphaGo (Silver et al., 2016), Alphafold (Senior et al., 2020) and GPT-3 (Brown et al., 2020). However, billions of resource-constrained mobile and IoT devices have become the primary data source to empower the intelligence of many applications (Bonawitz et al., 2019; Brisimi et al., 2018; Li et al., 2019a). Due to privacy, security, regulatory and economic considerations (Voigt & Von dem Bussche, 2017; Li et al., 2018), it is increasingly difficult and undesirable to pool data together for centralized training. Therefore, federated learning approaches (McMahan et al., 2017; Smith et al., 2017; Caldas et al., 2018; Kairouz et al., 2019; Yang et al., 2019) allow all the participants to reap the benefits of shared models without sharing private data have become increasingly attractive.

In a typical Federated Learning (FL) training process using FedAvg (McMahan et al., 2017), each client sends its model parameters or gradients to a central server, which aggregates all clients' updates and sends the aggregated parameters back to the clients to update their local model. Because FL places computation burden on edge devices, its learnability is largely limited by the edge resources, on which training large models is often impossible. On the other hand, the server only directly aggregates the clients' models and its computation power is not fully exploited. In this work, we investigate the paradigm where the server adopts a *larger* model in FL. Here we use *larger server model*, or GEM to denote the setting where the size of the server model is larger than that of the clients (See Table 1). Making FL trainable with GEM is desirable to break through model capacity and enable collaborative knowledge fusion and accumulation at server.

One feasible approach to bridge FL with GEM is through knowledge distillation (KD) (Hinton et al., 2015), where clients and the server transfer knowledge through logits. For example, FedGKT (He et al., 2020a) adopts a server model as a downstream sub-model and transfers knowledge directly from smaller edge models. In FedGKT the large server model essentially learns from one small teacher at a time and doesn't learn consensus knowledge from multiple teachers. In FL, KD has also been applied (Li & Wang, 2019; Lin et al., 2020) to transfer ensemble knowledge of clients through consensus of output logits rather than parameters. These works either assume no model on the server

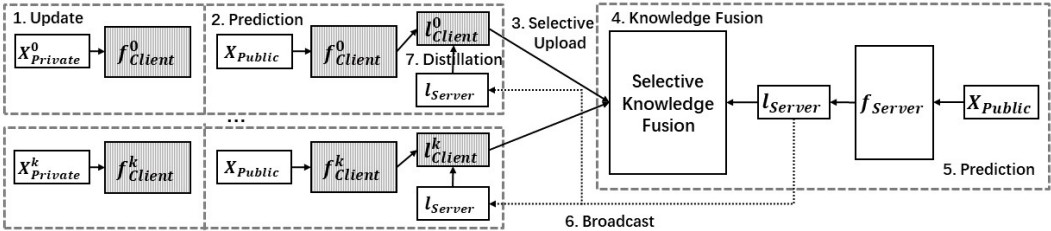

**Figure 1:** The framework of FedGEMS.

(Li & Wang, 2019) or a prototype server model of the same architecture as the client model (Lin et al., 2020).

**Contributions.** In this paper, we first propose a new paradigm to bridge FL with GEM, termed **FedGEM**, which can learn effectively and efficiently from fused knowledge by resource-constrained clients, and is also able to transfer knowledge back to clients with heterogeneous architectures. To further prevent negative and malicious knowledge transfer, we carefully design a selection and weighting criterion to enhance our knowledge transfer protocol, termed **FedGEMS**. We demonstrate with extensive experiments on various image classification tasks that our results significantly surpass the previous state-of-the-art baselines in both homogeneous and heterogeneous settings. Furthermore, thanks to our effective and selective protocol, our framework improves the robustness of FL on various malicious attacks and significantly reduces the overall communication. In summary, we propose a new framework to bridge FL with larger server models and simultaneously consolidate several benefits altogether, including superior performance, robustness of the whole system, and lower communication cost.

## 2 RELATED WORK

**Federated Learning with GEM.** FL is a collaborating learning framework without sharing private data among the clients. The classical method FedAvg (McMahan et al., 2017) and its recent variations (Mohri et al., 2019; Lin et al., 2018; Li et al., 2019b) directly transfer the clients' parameters or gradients to the server nodes. To tackle the performance bottleneck by training resource-constrained clients in FL, there are two lines of work to bridge FL with GEM. The first line of studies adopts model compression (Han et al., 2015; He et al., 2018; Yang et al., 2018), manually designed architectures (Howard et al., 2017; Zhang et al., 2018; Iandola et al., 2016) or even efficient neural architecture search (Tan & Le, 2019; Wu et al., 2019) to adapt a GEM to on-device learning. Another line is adopting knowledge distillation (Hinton et al., 2015) to transfer knowledge through output logits rather than parameters between a client model and a GEM (He et al., 2020a). However, FedGKT (He et al., 2020a)'s focus is to transfer knowledge directly from clients to server without considering the consensus knowledge fused from clients. Therefore its performance is limited.

**Federated Learning with Knowledge Distillation.** In fact, ensemble knowledge distillation has been shown to boost collaborative performance in FL. Specifically, FedMD (Li & Wang, 2019) adopts a labeled public dataset and averaged logits to transfer knowledge. FedDF (Lin et al., 2020) proposes ensemble distillation for model fusion by aggregating both logits and models from clients. In addition, KD is used to enhance robustness in FL. Cronus (Chang et al., 2019) and DS-FL (Itahara et al., 2020) utilize a public dataset with soft labels jointly with local private dataset for local training, and combine with Cronus or entropy reduction aggregation, respectively, to defend against poisoning attacks in FL. In this work, we propose to exploit the benefit of both a larger server model and client knowledge fusion. Our work is also motivated by the recent studies in KD (Qin et al., 2021; You et al., 2017; Li et al., 2021; Yuan et al., 2021; Wang et al., 2021), which show that student models can have larger capacities by learning from multiple teacher models.

| Method | Public Data | Client Model Heterogeneity | Aggregation | Server model Size |
|--------|-------------|----------------------------|-------------|-------------------|
| **FedAvg** | - | No | Average | - |
| **FedMD** | Labeled | Yes | Average | - |
| **Cronus** | Unlabeled | Yes | Cronus | - |
| **FedDF** | Unlabeled | No | Average | = Client |
| **FedGKT** | - | Yes | - | > Client |
| **DS-FL** | Unlabeled | Yes | Entropy-reduction | - |
| **FedGEM** | Labeled | Yes | Average | > Client |
| **FedGEMS** | Labeled | Yes | Selective | > Client |

**Table 1:** Comparison of FedGEMS with related works.

## 3 METHODOLOGY

### 3.1 PRELIMINARIES

We assume that there are $K$ clients in federated learning process. The $k$th client has its own private labeled dataset $\boldsymbol{X}^k := \{(\boldsymbol{x}_i^k, \boldsymbol{y}_i^k)\}_{i=1}^{N^k}$ that can be drawn from the same or different distribution, where $\boldsymbol{x}_i^k$ is the $i$th training sample in the $k$th client model, $\boldsymbol{y}_i^k$ is its corresponding ground truth label, and $N^k$ denotes the total number of samples. Each client also trains its own model $\boldsymbol{f}_c^k$ which can be of the same architecture (**homogeneous**) or different architecture (**heterogeneous**). There is also a public dataset $\boldsymbol{X}^0 := \{(\boldsymbol{x}_i^0, \boldsymbol{y}_i^0)\}_{i=1}^{N^0}$ which is accessible to both server and clients. On the server side, we assume that there is a larger server model to be trained, denoted as $\boldsymbol{f}_s$. $\boldsymbol{L}_s$ and $\boldsymbol{L}_c^k$ denotes the logit tensors from the server and the $k$th client model.

### 3.2 FEDGEMS FRAMEWORK

We illustrate our overall framework in Fig. 1 and summarize our training algorithm in Algorithm 1.

**FedGEM.** During each communication round, all client models first use private datasets to train several epochs, then transfer the predicted logits on public dataset as knowledge to the server model. The server model aggregates the clients' logits and then trains its server model with the guidance of fused knowledge. After training, the server model then transfers its logits back to all client models. Finally, each client model distills knowledge from received logits and continues their training on private datasets. Continuously iterating over multiple rounds, both the server and client models mutually learn knowledge from each other. Through this alternating training processing, we can obtain a large server model with accumulated knowledge and an ensemble of high-performance client models.

**FedGEMS**: In FedGEMS, the server adopts a selection and weighting criterion to select knowledgeable clients for aggregation, which is detailed in the next section 3.3.

To illustrate the features of our framework, we compare it with the related studies of KD-based methods in federated learning in Table 1 on the following aspects: whether they use a labeled, unlabeled or no public dataset, whether client models can have heterogeneous architectures, the aggregation strategy on the server, and whether the server has a larger model. Note FedGEM can be regarded as placing a larger server model on top of the FedMD framework while keeping other settings the same.

### 3.3 SELECTIVE KNOWLEDGE FUSION IN SERVER MODEL

Since clients' knowledge may negatively impact the server model in the heterogeneous or malicious setting and vice versa, we further propose selective strategies on both server and client sides to enforce positive knowledge fusion into the large server model as shown in Fig. 2.

**Algorithm 1 Illustration of the Framework of FedGEMS.** $T$ is the number of communication rounds; $\boldsymbol{X}^0$ and $\boldsymbol{Y}^0$ denotes the images and their corresponding labels in public dataset; $\boldsymbol{X}^k$ and $\boldsymbol{Y}^k$ denotes the private dataset in the $k$th client model; $\boldsymbol{f}_s$ and $\boldsymbol{f}_c^k$ are the server with parameter $\boldsymbol{W}_s$ and the $k$th client model with parameters $\boldsymbol{W}_c^k$; $\boldsymbol{L}_{\text{Global}}$ indicates the global logits to save correct logits; $\boldsymbol{L}_s$ and $\boldsymbol{L}_c^k$ are the logit tensors from the server and the $k$th client model.

```
 1: ServerExecute():                                          17: ClientTrain(Ls):
 2: for each round t = 1, 2, ..., T do                        18: for each client kth in parallel do
 3:     // Selective Knowldge Fusion                          19:     // Knowledge Distillation on Clients
 4:     for idx, x⁰, y⁰ ∈ {X⁰, Y⁰} do                         20:     for x⁰, y⁰ ∈ {X⁰, Ls, Y⁰} do
 5:         if fs(Ws; x⁰) == y⁰ then                          21:         Lc ← LC(x⁰, y⁰, Ls)        ▷ in Eq. 6
 6:             Ls ← LS₁(x⁰, y⁰)        ▷ in Eq. 1            22:         Wcᵏ ← Wcᵏ − ηk∇Lc
 7:             LGlobal[idx] ← Ls[idx]                         23:     // Local Training on Clients
 8:         else if idx in LGlobal then                       24:     for xᵏ, yᵏ ∈ {Xᵏ, Yᵏ} do
 9:             Ls ← LS₂(x⁰, y⁰, LGlobal)  ▷ in Eq. 2         25:         Lc ← LCE(xᵏ, yᵏ)
10:         else                                              26:         Wcᵏ ← Wcᵏ − ηk∇Lc
11:             Lc ← ClientSelect(idx)
12:             Ls ← LS₃(x⁰, y⁰, Lc)
                               ▷ in Eq. 3, 4, 5               27: ClientSelect(idx):
13:         Ws ← Ws − ηk∇Ls                                   28: // Selective Transfer to Server
14:         Ls[idx] ← fs(Ws; x⁰)                              29: for each client kth in parallel do
15:     // Transfer Knowledge to Client Models                30:     Lc⁰[idx] ← fcᵏ(Wcᵏ; x⁰[idx])
16:     ClientTrain(Ls)                                       31: Return Lc to server
```

### 3.3.1 SELF-DISTILLATION OF SERVER KNOWLEDGE

At each iteration, the server model first performs self evaluation on the public dataset and split the samples into two classes, those it can predict correctly, $S_{\text{Correct}}$, and those it predicts wrongly, $S_{\text{Incorrect}}$. For each sample $\boldsymbol{x}^i$ in $S_{\text{Correct}}$ where the model prediction matches the ground truth label, we simply adopt the cross-entropy loss between the predicted values and the ground truth labels to train the server model.

$$\mathcal{L}_{S_1} = \mathcal{L}_{CE} = -\boldsymbol{y}^i \log(\boldsymbol{f}_s(\boldsymbol{x}^i)) \tag{1}$$

At the same time, we save the correct logit $\boldsymbol{l}_s^i$ into the global pool of logits as $\boldsymbol{l}_{Global}^i$, which can be further used as memory to recover its reserved knowledge from self-distillation.

For each sample $\boldsymbol{x}^i$ that the server predicts wrongly, we first check whether its corresponding global logit $\boldsymbol{l}_{Global}^i$ exists or not. We denote all samples that do not exist in $\boldsymbol{l}_{Global}$ as $S_{\text{Incorrect}}^*$, for which we believe the server model can not learn the sample entirely by itself and propose to use clients' collective knowledge as its teacher, which will be explained in the next section. If $\boldsymbol{l}_{Global}^i$ exists, it means that the knowledge was reserved by the server before, the server model performs self-distillation to recover this part of knowledge. The final training objective of self-distillation can be formulated as follows, where $\mathcal{D}_{KL}$ is the Kullback Leibler (KL) Divergence function and $\epsilon$ is a hyper-parameter to control the weight of knowledge distillation in all the following formulations.

$$\mathcal{L}_{S_2} = \mathcal{L}_{SS} = \epsilon\mathcal{L}_{CE} + (1 - \epsilon)\mathcal{D}_{KL}\left(\boldsymbol{l}_{Global}^i \| \boldsymbol{l}_s^i\right) \tag{2}$$

Our self-distillation strategy has at least two advantages. On the one hand, distilling knowledge from itself is better than from other models which have different architectures. On the other hand, learning from its stored logits can greatly reduce the communication cost as compared to transferring all redundant logits to client models.

### 3.3.2 SELECTIVE ENSEMBLE DISTILLATION OF CLIENT KNOWLEDGE

For those samples in $S_{\text{Incorrect}}^*$ that the server model fails to predict correctly, we try to distill knowledge from the ensemble of clients. Furthermore, considering the correctness and relative importance of client models, we propose a weighted selective strategy on client side.

Given an instance $(\boldsymbol{x}_i, \boldsymbol{y}_i)$ in $S_{\text{Incorrect}}^*$, we first split all clients $\{C_1, C_2, ..., C_K\}$ into the reliable and unreliable clients according to their predictions $\boldsymbol{p}_{C_j}$. For those clients who predict the wrong labels, we consider them as unreliable and discard their knowledge by setting their weights equal to 0. As

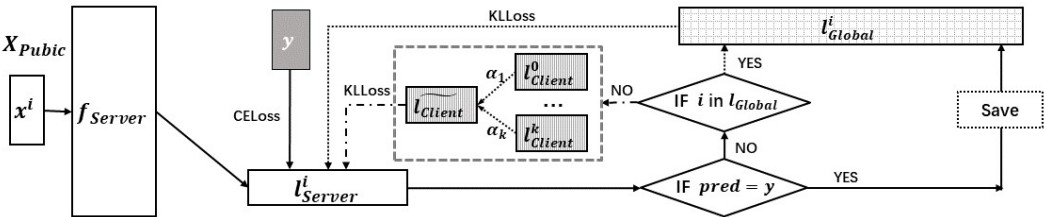

**Figure 2:** Selective knowledge fusion module.

for the rest of clients who predict the labels correctly, we consider them as reliable and use their entropy $H(\boldsymbol{p}_{C_j})$ as a measure of the confidence.

$$H(\boldsymbol{p}_{C_j}) = -\sum_{i=1}^{N} \boldsymbol{p}(\boldsymbol{x}_i) log \boldsymbol{p}(\boldsymbol{x}_i) \tag{3}$$

Following previous work (Li et al., 2021; Pereyra et al., 2017; Szegedy et al., 2016), low entropy indicates high confidence, and vice versa. Specifically, given an instance $(\boldsymbol{x}_i, \boldsymbol{y}_i)$ in $S^*_{\text{Incorrect}}$, we design its corresponding weights to aggregate output logits from different clients as below.

$$\boldsymbol{\alpha}_{C_j} = \begin{cases} 0, & C_j \in C_{\text{Unreliable}} \\ \text{softmax}(\frac{1}{H(\boldsymbol{p}_{C_j})}), & C_j \in C_{\text{Reliable}} \end{cases} \tag{4}$$

Therefore, for samples in $S^*_{\text{Incorrect}}$, the knowledge transferred from ensemble clients to server model can be formulated as the following.

$$\mathcal{L}_{S_3} = \mathcal{L}_{CS} = \epsilon\mathcal{L}_{CE} + (1-\epsilon)\mathcal{D}_{KL}\left(\sum_{j=1}^{K} \boldsymbol{\alpha}_{C_j} \boldsymbol{l}_{C_j} \Big\| \boldsymbol{l}_s\right) \tag{5}$$

Up to now, we have covered the selective knowledge fusion strategy for the server, next we will discuss the knowledge distillation on clients.

### 3.4 Training in Client Models

Each client model first receives the logits $\boldsymbol{l}^i_{Server}$ of public dataset from the server model. Then it distills knowledge according to the logits as well as computes the cross-entropy loss to train on public dataset.

$$\mathcal{L}_C = \epsilon\mathcal{L}_{CE} + (1-\epsilon)\mathcal{D}_{KL}\left(\boldsymbol{l}^i_s \| \boldsymbol{l}^i_c\right) \tag{6}$$

After knowledge distillation, each client model further adopts the cross-entropy loss to train on its local private dataset to better fit into its target distribution.

## 4 Experimental Evaluations

### 4.1 Experiment Settings

**Task and Dataset.** For fair comparison, we use the same training tasks as He et al. (2020a), which include image classifications on the CIFAR-10 (Krizhevsky et al., 2009) and CIFAR-100 (Krizhevsky et al., 2009) datasets. More details about these two datasets can be found in Appendix B.1. For each dataset, we randomly split data into a public and a private dataset with a ratio of 1:1. We further study the impact of this ratio in Sec. 5.2 of our experiments. The public dataset is used for transferring knowledge between server node and client nodes, while the private dataset is for client training. Both of them further split into a training and a testing dataset with a ratio of 5:1.

**Homogeneous Setting.** In our main experiments, we randomly shuffle and partition private dataset into 16 clients. As shown in Appendix B.6, our client models all adopt a tiny CNN architecture called ResNet-11, while the server model architecture is ResNet-56. To investigate the influence of server model size and client number, we further vary the server model size from ResNet-20 to ResNet110 and the total client number from 4 to 64 in Sec. 5.3 and Sec. 5.4, respectively.

**Heterogeneous Setting.** We adopt the Dirichlet distribution (Yurochkin et al., 2019; Hsu et al., 2019) to control the degree of heterogeneity in our non-iid setting. In this experiment, our $\alpha$ is 0.5. We also introduce model heterogeneity in our experiments. The client models vary from ResNet-11 to Resnet-17 as shown in Appendix B.7.

**Implementation Details.** In both homogeneous and heterogeneous settings, the number of local epochs are set to 1 after careful tuning, consistent with implementations in FedGKT. The number of communication round to reach convergence is 400. We adopt Adam optimizer (Kingma & Ba, 2014) with an initial learning rate 0.001. Following He et al. (2020a), the learning rate can reduce once the accuracy is stable (Li & Arora, 2019). As for the factor $\epsilon$, we find that the best choice is 0.75. Its impacts on server and client performance are shown in Appendix B.5 .

**Baselines.** We compare our framework FedGEM/FedGEMS with the following baselines. (1) **Stand-Alone**: the server and client models train on their local datasets $X^0$ and $X^k$, respectively; (2) **Centralized**: the client model trains on the whole private dataset $\{X^1 + \cdots + X^k\}$; (3) **Centralized-All**: the server and clients perform centralized training on the whole dataset $\{X^0 + \cdots + X^k\}$, respectively. Note in reality no party has this complete dataset so this can be viewed as an upper limit of model performance. In addition, we also compare our performance with several existing related works, including **FedAvg** (McMahan et al., 2017), **FedMD** (Li & Wang, 2019), **Cronus** (Chang et al., 2019), **DS-FL** (Itahara et al., 2020), **FedDF** (Lin et al., 2020) and **FedGKT** (He et al., 2020a). For fair comparison, the settings of client models and the split of public and private dataset in all approaches are kept the same. As for the server model, FedDF adopts the same architecture as its clients' model, ResNet-11, according to its design, while FedGKT and our FedGEMS utilize ResNet-56. Note FedGKT (He et al., 2020a) only reported the performance of the server model. We empirically found that its clients' performance is lower due to constraints of model size. For a fair comparison with FedGKT, we only reported its performance on server side. More details of their important hyper-parameters in our experiments are listed in Appendixes B.9, respectively.

## 4.2 PERFORMANCE EVALUATIONS

| Method | Homo | | | | Hetero | | | |
|---|---|---|---|---|---|---|---|---|
| | CIFAR-10 | | CIFAR-100 | | CIFAR-10 | | CIFAR-100 | |
| | Server | Clients | Server | Clients | Server | Clients | Server | Clients |
| **Stand-Alone** | 84.03 | 41.57 | 65.38 | 26.95 | 84.03 | 28.58 | 65.38 | 18.70 |
| **Centralized** | - | 76.21 | - | 58.61 | - | 83.54 | - | 63.44 |
| **Centralized-All** | 91.27 | 82.45 | 78.13 | 65.51 | 91.27 | 88.54 | 78.13 | 70.96 |
| **FedAvg** | - | 53.33 | - | 31.47 | - | - | - | - |
| **FedMD** | - | 58.47 | - | 34.77 | - | 51.11 | - | 25.57 |
| **Cronus** | - | 57.73 | - | 34.19 | - | 53.47 | - | 29.71 |
| **DS-FL** | - | 50.78 | - | 25.70 | - | 43.83 | - | 16.69 |
| **FedDF** | - | 56.31 | - | 31.27 | - | - | - | - |
| **FedGKT** | 55.67 | - | 29.89 | - | 45.55 | - | 26.96 | - |
| **FedGEM** | 86.62 | 80.35 | 67.72 | 62.61 | 87.11 | 80.14 | 67.27 | 64.73 |
| **FedGEMS** | **88.08** | **81.86** | **69.08** | **63.81** | **87.97** | **84.18** | **67.72** | **65.93** |

**Table 2:** Model performance in homogeneous and heterogeneous settings.

Table 2 shows the comparisons of our approaches with other baselines with ResNet-56 as the server model. First of all, our approach FedGEM outperforms all the above KD-based federated learning baselines significantly on both server and client side, demonstrating the effectiveness of knowledge transfer with a larger server model. Compared with FedGKT which also adopts a larger server model, our performance is significantly improved by knowledge fusion from multiple clients. Compared to the performance of stand-alone server and clients, our framework simultaneously improves both the sever model and client model, indicating that server and clients mutually benefit from knowledge transfer. Using public labels for supervision and selection, our FedGEMS framework is able to enhance positive knowledge fusion and further improve performance on both server and clients. We also conduct experiments with 8 clients and ResNet-20 as server model, shown in Appendix B.4.

## 4.3 ROBUSTNESS

**Poisoning Attacks.** Following Chang et al. (2019), we employ model poisoning attacks to evaluate the robustness of our methods. Our model poisoning attacks include Naive Poisioning (PAF), Little Is Enough Attack (LIE) (Baruch et al., 2019) and OFOM (Chang et al., 2019). In PAF and LIE, the attacker poisons one client at each round via disturbing the logits or parameters which transfer from clients to server. In OFOM, two clients' benign predictions or parameters are poisoned at each round. The detailed algorithms about different poisoning attacks can be found in Appendix A.

| Method | PAF | | LIE | | OFOM | |
|---|---|---|---|---|---|---|
| | Server | Client | Server | Client | Server | Client |
| **FedMD** | - | 16.13 /-68.44 | - | 16.21/-68.28 | - | 22.53/-55.92 |
| **DS-FL** | - | 37.94/-13.44 | - | 23.42/-46.57 | - | 23.32/-46.79 |
| **Cronus** | - | **53.71/+00.45** | - | **53.98/+00.94** | - | **53.74 /-00.50** |
| **FedGKT** | 13.82/-69.66 | - | 10.42/-77.12 | - | 14.20/-68.83 | - |
| **FedGEM** | 55.94/-35.78 | 72.78/-09.18 | 74.86/-14.06 | 74.32/-07.26 | 85.22/-02.17 | 74.47/-07.08 |
| **FedGEMS** | **87.46/-05.80** | **82.58/-01.90** | **87.95/-00.02** | **83.27/-01.08** | **88.53/+00.64** | **82.76/-01.67** |

**Table 3:** Comparison of model poisoning attacks in heterogeneous setting. The A in "A/B" denotes the model performance after attacks, while B denotes the percentage changes compared to the original accuracy ("+" denotes increasing, and "-" denotes dropping). We remark both the best performance and the least changed ratio (%) to its original accuracy.

The results of poisoning attacks in heterogeneous setting on CIFAR-10 dataset are shown in Table 3. By adopting a selective strategy, our proposed FedGEMS provides consistent robustness across various attacks on both server and client models. FedGEM demonstrates superior robustness to FedMD on various model poisoning attacks. DS-FL is relatively more robust than FedMD via its proposed entropy reduction aggregation based on FedMD framework. Since Cronus specifically designs secure robust aggregation in a FedMD framework, its robustness to the attacks is strong, as expected. The performance of FedGKT drops marginally because it transmits extra feature maps which is vulnerable. Our selective knowledge transfer strategy shows comparably strong robustness on both clients and server side without additional computation cost for robust aggregation algorithms.

## 4.4 COMMUNICATION COST

Following Itahara et al. (2020), we further evaluate the cumulative communication costs required to achieve specific accuracy in heterogeneous setting on CIFAR-10. The epochs in both client and server sides among all methods to achieve their best performances are the same as 1 which further ensures fairness. The formulations to calculate communication cost for different methods are shown in Appendix B.2.

| Method | ComU@40% (MB) | ComU@50% (MB) | ComU@80% (MB) | Top-Acc (%) |
|---|---|---|---|---|
| **FedMD** | 976.56 | 5,493.16 | - | 51.11 |
| **Cronus** | 610.35 | 2,471.92 | - | 53.47 |
| **DS-FL** | 3,692.63 | - | - | 43.83 |
| **FedGKT** | 31,250.00 | - | - | 45.55 |
| **FedGEM** | 91.55 | 244.14 | 3,261.27 | 80.14 |
| **FedGEMS** | 23.38 | 190.87 | 1,894.84 | 84.18 |

**Table 4:** Comparison of communication cost and Top-Accuracy. ComU@x: the cumulative communication cost required to achieve the absolute accuracy of x. Top-Accuracy: the highest testing accuracy among the training process.

The results are shown in Table 4. It is worth noting that FedMD, Cronus, DS-FL and FedGEM cost the same per round according to their designs. Therefore, to achieve the accuracy of 45%, FedGEM costs much lower than other three methods which indicates that FedGEM converges fast. The communication cost of FedGKT is much higher than others because it sends a feature map per private data in training. Specifically, the communication cost per feature map is 64kb while the cost per logit is only 0.039kb. Furthermore, the communication cost of FedGEMS keeps lower than FedGEM in the whole training phase which means that our selective strategy can effectively reduce uploads to achieve the same performance.

# 5 UNDERSTANDING FEDGEMS

## 5.1 KNOWLEDGE ACCUMULATION AT SERVER

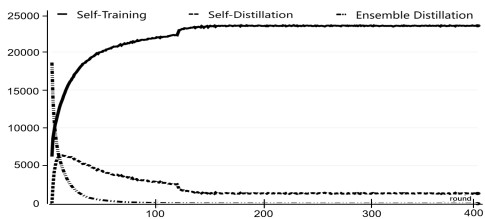

| Method | Server | Client |
|---|---|---|
| FedGEMS | **88.08** | **81.86** |
| - Self-Training | 85.73 | 80.17 |
| - Self-Distillation | 86.12 | 81.30 |
| - Ensemble Distillation | 86.55 | 80.01 |

**Figure 3:** Total number of samples in different selective decisions.

**Table 5:** Ablation studies of three components in selective strategy.

To analyze the selective knowledge fusion process of the server model in our framework, we report the number of samples associated with each step in our decision-making process in our experiment on CIFAR-10 with 25,000 samples as public dataset. Each sample will choose one of three strategies according to our selective knowledge fusion module. Specifically, in Fig. 3, the line of self-training indicates the total number of samples the server model predicts correctly, while the lines of self-distillation and ensemble-distillation indicate the number of samples the server learns via self-distillation and client-side ensemble knowledge, respectively. In the initial stage, since both the server and client models learn from scratch, the number of samples that the server model can predict correctly is limited, and most of the knowledge is accumulated by distillation from the client models' fusion. As the training progresses, the server model needs to restore some of its knowledge from a self-distillation strategy. With the server model continually fusing sufficient knowledge, the model performance of server model eventually exceeds client models and the number of transferred samples from clients dropped to 0. Notice that in the final stage, there still remains some stubborn samples which are hard for the server model while most samples can be solved by the large server model with accumulated knowledge.

Furthermore, we conduct a series of ablation studies to detect the contributions of different components to the accumulated knowledge at server side. The results are shown in Table 5 and all the performances of both server and client models decrease. This phenomenon indicates that each component contributes to the overall performance. Comprehensively, self-training is the most important component and the most probable reason is that learning from the model itself is the most effective way. Ensemble distillation which avoids transferring knowledge from malicious clients can also help prompt the performance of client models in return.

## 5.2 EFFECT OF PUBLIC AND PRIVATE DATASET RATIO

Concerning the size of public dataset, we evaluate the effect of public and private dataset ratio on both model performance and communication cost per round. To change the ratio, we fix the size of private dataset, and vary the size of public dataset from 5,000 to 25,000.

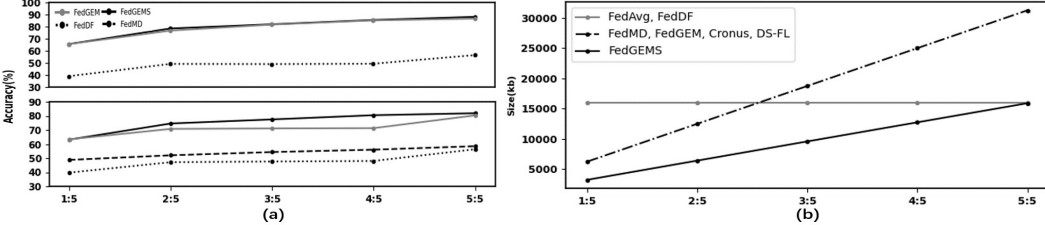

**Figure 4:** (a) Model performance of both server (top) and client (bottom) models on public dataset of different sizes; (b) Communication cost per round on public dataset of different sizes.

The results of both model performance and communication cost per round are shown in Fig. 4 (a) and (b) respectively. As for model performance, it can be seen that both the client and server model continuously improve as the size of public data increases, indicating that their performances are highly correlated. Our performance consistently outperforms FedMD, and the performance gap over

FedMD continuously grows as more public dataset is available. Note again FedGEM is essentially FedMD with a larger server model so the performance gain is due to the accumulated knowledge on the server side. As for FedDF which owns an ensemble server model as same architecture as clients, the performance in both server and client models remains a huge gap to our performance. This phenomenon further demonstrates the importance of the large capacity of server model. FedGEMS can still outperform FedGEM especially on the client side which indicates that our selective strategy is efficient to fuse positive knowledge to boost performance.

Due to the compact models deployed in client nodes, the bits of model parameters are limited, thus the communication overhead of FedAvg and FedDF is not always higher than typical KD-based methods when a large public dataset is used. Due to the enormous communication overhead of feature maps, we do not figure the flat line (irrelevance to the size of public dataset) of FedGKT which is always 1.6M (kb) per round. Thanks to our selection strategy, the communication cost of our proposed framework FedGEMS is the lowest among all methods. In summary, the results demonstrate that our selection strategy can greatly save the communication cost with reasonable public data size.

## 5.3 Effect of Server Model Size

We further investigate the influence of the server model size on the overall performance by gradually changing the server model from ResNet-20 to ResNet-110, while fixing the client models as ResNet-11. The details of the model parameters can be found in Appendix B.8.

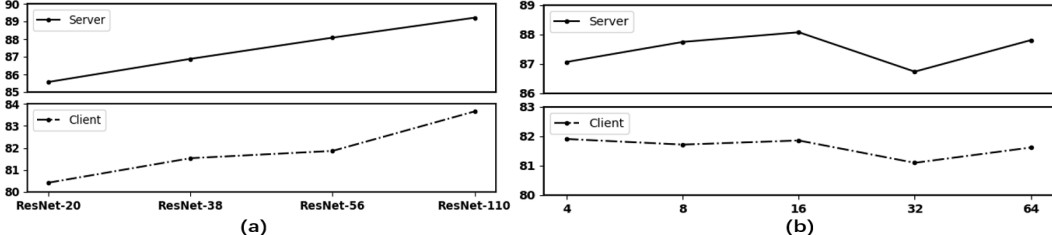

**Figure 5:** (a) Server (top) and client (bottom) model performances of different server model sizes; (b) server (top) and client (bottom) model performances of different numbers of clients.

The results are shown in Fig. 5 (a). It can be witnessed that the performance of both the server and client models improves as the server model becomes larger and deeper. This phenomenon verifies the importance of placing a larger deeper server model to boost the performance of both server and the resource-constrained client models.

## 5.4 Effect of Number of Clients

We vary the number of clients from 4 to 64 of our proposed method FedGEMS while keeping the total number of private samples the same. Thus the number of private samples per client vary accordingly. The experimental results shown in Fig. 5 (b) indicate that with increased number of clients and less client data, the performance of our approaches is basically stable especially in client side, benefiting from our larger server model setting with fused knowledge.

## 6 Conclusion

In this work, we first propose a new paradigm to apply a large deeper server model to effectively and efficiently fuse and accumulate knowledge, which can enhance the model performances on both server and client sides. Furthermore, we design a selection and weighted criterion on both sides to distill only positive knowledge into the server. We conduct a series of experiments to evaluate our proposed framework FedGEMS and our results show that FedGEMS can significantly surpass all baselines in both homogeneous and heterogeneous settings. Meanwhile, FedGEMS can further improve the robustness of FL on poisoning attacks as well as reduce the communication costs between server and client sides. Our framework has certain limitations, such as the dependence of a labeled public dataset. In future work, we will study the effectiveness of our work in other critical tasks, such as NLP and knowledge graph-based tasks, and in the settings where public and private data are from different domains or distributions.

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

## A  POISONING ATTACKS IN FEDERATED LEARNING

In this section, we introduce a set of poisoning attacks from the literature which used to evaluate the robustness in federated learning.

### A.1  NAIVE POISONING (PAF)

Naive poisoning is a type of model poisoning attack. The adversary is able to poison $\epsilon$ fraction of total client. With the knowledge of the distribution of benign updates, the opponent can introduce malicious update $\theta_m$. This attack is far from the mean of the benign updates, obtained by adding a significantly large vector $\theta'$ to it:

$$\theta_m = \frac{\sum_{i=1}^{n} \theta_i}{(1 - \epsilon)n} + \theta' \tag{7}$$

Then, the server updates with the malicious update $\theta_m$ and transfers it to all the clients.

### A.2  LITTLE IS ENOUGH ATTACK (LIE)

LIE is a type of model poisoning attack proposed by Baruch et al. (2019). The detailed implementation of LIE is summarized in Algorithm 2. The malicious update $\theta_m$ is achieved by making small changes in many dimensions to a benign update. The malicious update is trained in server and then shared by each client.

---

**Algorithm 2 Little is enough attack (LIE)** Baruch et al. (2019)

---

1: **Input:** The number of benign updates $n$, the fraction of malicious updates $\epsilon$
2: The number of majority benign clients:

$$s = \left\lfloor \frac{n}{2} + 1 \right\rfloor - \epsilon n \tag{8}$$

3: Using z-table, set:

$$z^{\max} = \max_z \left( \phi(z) < \frac{n-s}{n} \right) \tag{9}$$

4: **for** $j \in [d]$ **do**
5:     compute mean ($\mu_j$) and standard deviation ($\sigma_j$) of benign updates.

$$\theta_m^j \leftarrow \mu_j + z^{\max} \cdot \sigma_j \tag{10}$$

6: **Output:** $\theta_m$

---

### A.3 ONE FAR ONE MEAN (OFOM)

This model poisoning attack, OFOM, is proposed by Chang et al. (2019). In this attack, two malicious updates will be added to the benign updates. First, the opponent adds a significantly large vector $\theta'$ to the mean of the benign updates to obtain $\theta_m^1$. Then, the opponent crafts $\theta_m^2$, which is the mean of the benign updates and $\theta_m^1$.

$$\theta_1^m = \frac{\sum_{i=1}^n \theta_i}{n} + \theta', \quad \theta_2^m = \frac{\sum_{i=1}^n \theta_i + \theta_1^m}{n+1} \tag{11}$$

## B EXTRA EXPERIMENTAL RESULTS AND DETAILS

### B.1 A SUMMARY OF DATASET

Following He et al. (2020a), our experiments utilize CIFAR-10 (Krizhevsky et al., 2009) and CIFAR-100 (Krizhevsky et al., 2009) as our datasets. **CIFAR-10** consists 10 classes colour images, with 6000 images per class. **CIFAR-100** is a more challenging dataset with 100 subclasses that falls under 20 superclasses, e.g. *baby, boy, girl, man* and *woman* belong to *people*.

### B.2 FORMULATIONS OF COMMUNICATION COST

In the Fig. 6, we list the formulations to compute communication costs between server and client models in different methods.

| Method | Formulation |
|---|---|
| **FedAvg** | $\left( B_{paras} \right) \times (\#C + \#S)$ |
| **FedMD** **DS-FL** **Cronus** **FedGEM** | $\left( B_{logits} \right) \times (\#C + \#S)$ |
| **FedDF** | $\left( B_{logits} \right) \times (\#C) + \left( B_{paras} \right) \times (\#C + \#S)$ |
| **FedGKT** | $\left( B_{logits} \right) \times (\#C + \#S) + \left( B_{Features} \right) \times (\#C)$ |
| **FedGEMS** | $\left( B_{logits} \right) \times (\#C_{Selective} + \#S)$ |

**Figure 6:** Formulations of Communication Cost. $B$ indicates the corresponding bits of logits or parameters. $S$ and $C$ denotes Server and Client, while $\#S$ and $\#C$ denotes the total number of Server and Clients.

## B.3 Communication cost of the uploading logits

The results of the communication cost of the uploading logits per round are shown in Table 6.

| Method | 1:5 | 2:5 | 3:5 | 4:5 | 5:5 |
|---|---|---|---|---|---|
| **FedMD Cronus DS-FL FedGEM** | 3125.0 | 6250.0 | 9375.0 | 12500.0 | 15625.0 |
| **FedGEMS** | 105.8 | 155.4 | 200.8 | 233.6 | 266.3 |

**Table 6:** Communication cost of the logits uploading from clients.

## B.4 Results of 8 Clients and ResNet-20

We also conduct different setting to validate the stability of our proposed framework FedGEMS. In this series of experiments, we adopt 8 clients and set the server model as ResNet-20, while keeping other experiment setting the same as main paper.

### B.4.1 Performance Evaluations

The results of our performance are shown in Table 7.

| Method | Homo | | | | Hetero | | | |
|---|---|---|---|---|---|---|---|---|
| | CIFAR-10 | | CIFAR-100 | | CIFAR-10 | | CIFAR-100 | |
| | Server | Clients | Server | Clients | Server | Clients | Server | Clients |
| **Stand-Alone** | 81.60 | 57.38 | 61.90 | 33.78 | 81.60 | 44.55 | 61.90 | 27.26 |
| **Centralized** | - | 74.81 | - | 56.42 | - | 79.95 | - | 60.52 |
| **Centralized-All** | 88.97 | 79.77 | 73.51 | 65.07 | 88.97 | 87.96 | 73.51 | 71.75 |
| **FedAvg** | - | 63.64 | - | 39.16 | - | - | - | - |
| **FedMD** | - | 67.53 | - | 46.26 | - | 43.87 | - | 34.11 |
| **Cronus** | - | 66.03 | - | 45.00 | - | 39.61 | - | 38.38 |
| **DS-FL** | - | 60.36 | - | 47.63 | - | 36.46 | - | 31.05 |
| **FedDF** | 65.18 | 64.78 | 42.11 | 40.87 | - | - | - | - |
| **FedGKT** | 70.80 | 44.75 | 31.18 | 27.72 | 35.47 | 12.78 | 35.53 | 21.05 |
| **FedGEM** | 83.53 | 79.70 | 66.45 | 61.28 | 83.47 | 78.89 | 65.77 | 57.21 |
| **FedGEMS** | **85.65** | **81.10** | **67.31** | **61.70** | **85.32** | **80.38** | **65.91** | **58.61** |

**Table 7:** Model performance in homogeneous and heterogeneous settings.

### B.4.2 Robustness

The results of our robustness experiment are shown in Table 8.

### B.4.3 Communication cost

We further evaluate the communication costs per round by varying public dataset sizes for various approaches on CIFAR-10, and the results are shown in Fig. 7.

## B.5 Results of different $\epsilon$

The results of different $\epsilon$, which is a hyper-parameter to control the weight of knowledge distillation, are shown in Table 9.

| Method | PAF | | LIE | | OFOM | |
|---|---|---|---|---|---|---|
| | **Server** | **Clients** | **Server** | **Clients** | **Server** | **Clients** |
| **FedAvg** | - | 14.42/-49.22 | - | 09.50/-54.14 | - | 13.82/-49.82 |
| **FedDF** | 13.12/-52.06 | 13.97/-50.81 | 15.89/-49.29 | 25.82/-38.96 | 12.64/-52.54 | 14.40/-50.38 |
| **FedMD** | - | 33.27/-34.26 | - | 43.97/-23.56 | - | 32.74/-34.79 |
| **DS-FL** | - | 46.62/-13.74 | - | 44.74/-15.62 | - | 45.20/-15.16 |
| **Cronus** | - | **66.29/+00.26** | - | **66.08/+00.05** | - | **65.98/-00.05** |
| **FedGEM** | 60.01/-23.52 | 74.79/-04.91 | 43.62/-39.91 | 69.26/-10.44 | 83.71/+00.18 | 79.36/-00.34 |
| **FedGEMS** | **86.86/+01.21** | **80.85/-00.25** | **84.64/-01.01** | **80.48/-00.62** | **86.33/+00.68** | **80.78/-00.32** |

**Table 8:** Comparison of model poisoning attacks in homogeneous setting. The A in "A/B" denotes the model performance after attacks, while B denotes the percentage changes compared to the original accuracy ("+" denotes increasing, and "-"denotes dropping). We remark both the best performance and the least changed ratio (%) to its original accuracy.

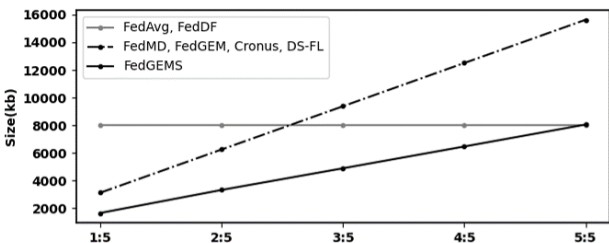

**Figure 7:** Communication cost per round of different public dataset sizes on CIFAR-10. Due to the enormous communication overhead of feature maps, we do not figure the flat line (irrelevance to the size of public dataset) of FedGKT which is always 1.6M (kb).

| Method | $\epsilon = 0.10$ | | $\epsilon = 0.5$ | | $\epsilon = 0.75$ | | $\epsilon = 0.90$ | |
|---|---|---|---|---|---|---|---|---|
| | **Server** | **Client** | **Server** | **Client** | **Server** | **Client** | **Server** | **Client** |
| **FedGEMS** | 88.71 | 80.37 | 87.78 | 81.97 | 88.08 | 81.86 | 84.50 | 79.53 |

**Table 9:** Comparison of model performance in different $\epsilon$.

## B.6 MODEL ARCHITECTURES OF HOMOGENEOUS SETTING

The model architectures of client and server models in homogeneous setting are shown in Table 10.

| Model | Conv1 | Conv2_x | | Conv3_x | | Conv4_x | | | Parameters |
|---|---|---|---|---|---|---|---|---|---|
| **ResNet-11** (Client) | $3 \times 3$ 16 Stride 1 | $\begin{bmatrix} 1 \times 1, 16 \\ 3 \times 3, 16 \\ 1 \times 1, 64 \end{bmatrix}$ | $\times 1$ | $\begin{bmatrix} 1 \times 1, 32 \\ 3 \times 3, 32 \\ 1 \times 1, 128 \end{bmatrix}$ | $\times 1$ | $\begin{bmatrix} 1 \times 1, 64 \\ 3 \times 3, 64 \\ 1 \times 1, 256 \end{bmatrix}$ | $\times 1$ | average pool 10-d fc | 127642 |
| **ResNet-20** (Server) | $3 \times 3$ 16 Stride 1 | $\begin{bmatrix} 1 \times 1, 16 \\ 3 \times 3, 16 \\ 1 \times 1, 64 \end{bmatrix}$ | $\times 2$ | $\begin{bmatrix} 1 \times 1, 32 \\ 3 \times 3, 32 \\ 1 \times 1, 128 \end{bmatrix}$ | $\times 2$ | $\begin{bmatrix} 1 \times 1, 64 \\ 3 \times 3, 64 \\ 1 \times 1, 256 \end{bmatrix}$ | $\times 2$ | average pool 10-d fc | 220378 |
| **ResNet-56** (Server) | $3 \times 3$ 16 Stride 1 | $\begin{bmatrix} 1 \times 1, 16 \\ 3 \times 3, 16 \\ 1 \times 1, 64 \end{bmatrix}$ | $\times 6$ | $\begin{bmatrix} 1 \times 1, 32 \\ 3 \times 3, 32 \\ 1 \times 1, 128 \end{bmatrix}$ | $\times 6$ | $\begin{bmatrix} 1 \times 1, 64 \\ 3 \times 3, 64 \\ 1 \times 1, 256 \end{bmatrix}$ | $\times 6$ | average pool 10-d fc | 591322 |

**Table 10:** Details of Model Architectures in Homogeneous Setting

## B.7 MODEL ARCHITECTURE OF HETEROGENEOUS SETTING

The different model architectures of client models in heterogeneous setting are shown in Table 11. The server model in heterogeneous setting is the same as homogeneous setting in Table 10.

| Model | Conv1 | Conv2_x | | Conv3_x | | Conv4_x | | |
|---|---|---|---|---|---|---|---|---|
| **ResNet-11(1, 9)** | $3 \times 3, 16$, stride 1 | $\begin{bmatrix} 1 \times 1, 16 \\ 3 \times 3, 16 \\ 1 \times 1, 64 \end{bmatrix}$ | $\times 1$ | $\begin{bmatrix} 1 \times 1, 32 \\ 3 \times 3, 32 \\ 1 \times 1, 128 \end{bmatrix}$ | $\times 1$ | $\begin{bmatrix} 1 \times 1, 64 \\ 3 \times 3, 64 \\ 1 \times 1, 256 \end{bmatrix}$ | $\times 1$ | average pool, 10-d fc |
| **ResNet-14(2, 10)** | $3 \times 3, 16$, stride 1 | $\begin{bmatrix} 1 \times 1, 16 \\ 3 \times 3, 16 \\ 1 \times 1, 64 \end{bmatrix}$ | $\times 1$ | $\begin{bmatrix} 1 \times 1, 32 \\ 3 \times 3, 32 \\ 1 \times 1, 128 \end{bmatrix}$ | $\times 1$ | $\begin{bmatrix} 1 \times 1, 64 \\ 3 \times 3, 64 \\ 1 \times 1, 256 \end{bmatrix}$ | $\times 2$ | average pool, 10-d fc |
| **ResNet-14(3, 11)** | $3 \times 3, 16$, stride 1 | $\begin{bmatrix} 1 \times 1, 16 \\ 3 \times 3, 16 \\ 1 \times 1, 64 \end{bmatrix}$ | $\times 1$ | $\begin{bmatrix} 1 \times 1, 32 \\ 3 \times 3, 32 \\ 1 \times 1, 128 \end{bmatrix}$ | $\times 2$ | $\begin{bmatrix} 1 \times 1, 64 \\ 3 \times 3, 64 \\ 1 \times 1, 256 \end{bmatrix}$ | $\times 1$ | average pool, 10-d fc |
| **ResNet-14(4, 12)** | $3 \times 3, 16$, stride 1 | $\begin{bmatrix} 1 \times 1, 16 \\ 3 \times 3, 16 \\ 1 \times 1, 64 \end{bmatrix}$ | $\times 2$ | $\begin{bmatrix} 1 \times 1, 32 \\ 3 \times 3, 32 \\ 1 \times 1, 128 \end{bmatrix}$ | $\times 1$ | $\begin{bmatrix} 1 \times 1, 64 \\ 3 \times 3, 64 \\ 1 \times 1, 256 \end{bmatrix}$ | $\times 1$ | average pool, 10-d fc |
| **ResNet-17(5, 13)** | $3 \times 3, 16$, stride 1 | $\begin{bmatrix} 1 \times 1, 16 \\ 3 \times 3, 16 \\ 1 \times 1, 64 \end{bmatrix}$ | $\times 1$ | $\begin{bmatrix} 1 \times 1, 32 \\ 3 \times 3, 32 \\ 1 \times 1, 128 \end{bmatrix}$ | $\times 2$ | $\begin{bmatrix} 1 \times 1, 64 \\ 3 \times 3, 64 \\ 1 \times 1, 256 \end{bmatrix}$ | $\times 2$ | average pool, 10-d fc |
| **ResNet-17(6, 14)** | $3 \times 3, 16$, stride 1 | $\begin{bmatrix} 1 \times 1, 16 \\ 3 \times 3, 16 \\ 1 \times 1, 64 \end{bmatrix}$ | $\times 2$ | $\begin{bmatrix} 1 \times 1, 32 \\ 3 \times 3, 32 \\ 1 \times 1, 128 \end{bmatrix}$ | $\times 1$ | $\begin{bmatrix} 1 \times 1, 64 \\ 3 \times 3, 64 \\ 1 \times 1, 256 \end{bmatrix}$ | $\times 2$ | average pool, 10-d fc |
| **ResNet-17(7, 15)** | $3 \times 3, 16$, stride 1 | $\begin{bmatrix} 1 \times 1, 16 \\ 3 \times 3, 16 \\ 1 \times 1, 64 \end{bmatrix}$ | $\times 2$ | $\begin{bmatrix} 1 \times 1, 32 \\ 3 \times 3, 32 \\ 1 \times 1, 128 \end{bmatrix}$ | $\times 2$ | $\begin{bmatrix} 1 \times 1, 64 \\ 3 \times 3, 64 \\ 1 \times 1, 256 \end{bmatrix}$ | $\times 1$ | average pool, 10-d fc |
| **ResNet-17(8, 16)** | $3 \times 3, 16$, stride 1 | $\begin{bmatrix} 1 \times 1, 16 \\ 3 \times 3, 16 \\ 1 \times 1, 64 \end{bmatrix}$ | $\times 1$ | $\begin{bmatrix} 1 \times 1, 32 \\ 3 \times 3, 32 \\ 1 \times 1, 128 \end{bmatrix}$ | $\times 1$ | $\begin{bmatrix} 1 \times 1, 64 \\ 3 \times 3, 64 \\ 1 \times 1, 256 \end{bmatrix}$ | $\times 2$ | average pool, 10-d fc |

**Table 11:** Details of the 16 client models architecture used in our experiment

## B.8 MODEL ARCHITECTURES OF LARGER SERVER MODELS

The detailed model architectures of large server models are shown in Table 12.

| Model | Conv1 | Conv2_x | | Conv3_x | | Conv4_x | | |
|---|---|---|---|---|---|---|---|---|
| **ResNet-20** | $3 \times 3, 16$, stride 1 | $\begin{bmatrix} 1 \times 1, 16 \\ 3 \times 3, 16 \\ 1 \times 1, 64 \end{bmatrix}$ | $\times 2$ | $\begin{bmatrix} 1 \times 1, 32 \\ 3 \times 3, 32 \\ 1 \times 1, 128 \end{bmatrix}$ | $\times 2$ | $\begin{bmatrix} 1 \times 1, 64 \\ 3 \times 3, 64 \\ 1 \times 1, 256 \end{bmatrix}$ | $\times 2$ | average pool, 10-d fc |
| **ResNet-38** | $3 \times 3, 16$, stride 1 | $\begin{bmatrix} 1 \times 1, 16 \\ 3 \times 3, 16 \\ 1 \times 1, 64 \end{bmatrix}$ | $\times 4$ | $\begin{bmatrix} 1 \times 1, 32 \\ 3 \times 3, 32 \\ 1 \times 1, 128 \end{bmatrix}$ | $\times 4$ | $\begin{bmatrix} 1 \times 1, 64 \\ 3 \times 3, 64 \\ 1 \times 1, 256 \end{bmatrix}$ | $\times 4$ | average pool, 10-d fc |
| **ResNet-56** | $3 \times 3, 16$, stride 1 | $\begin{bmatrix} 1 \times 1, 16 \\ 3 \times 3, 16 \\ 1 \times 1, 64 \end{bmatrix}$ | $\times 6$ | $\begin{bmatrix} 1 \times 1, 32 \\ 3 \times 3, 32 \\ 1 \times 1, 128 \end{bmatrix}$ | $\times 6$ | $\begin{bmatrix} 1 \times 1, 64 \\ 3 \times 3, 64 \\ 1 \times 1, 256 \end{bmatrix}$ | $\times 6$ | average pool, 10-d fc |
| **ResNet-110** | $3 \times 3, 16$, stride 1 | $\begin{bmatrix} 1 \times 1, 16 \\ 3 \times 3, 16 \\ 1 \times 1, 64 \end{bmatrix}$ | $\times 12$ | $\begin{bmatrix} 1 \times 1, 32 \\ 3 \times 3, 32 \\ 1 \times 1, 128 \end{bmatrix}$ | $\times 12$ | $\begin{bmatrix} 1 \times 1, 64 \\ 3 \times 3, 64 \\ 1 \times 1, 256 \end{bmatrix}$ | $\times 12$ | average pool, 10-d fc |

**Table 12:** Details of the large server models architecture used in our experiment

## B.9 HYPER-PARAMETERS

In table 13, we sum up the hyper-parameter settings for all the methods in our experiments. We directly run FedAvg, FedGKT and re-implement all the other methods base on an open-source federated learning research library FedML (He et al., 2020b) in a distributed computing environment.

| Methods | Hyperparameters | Homogeneous | Heterogeneous |
|---|---|---|---|
| **FedGEMS** | optimizer
batch size
client epochs of public data
client epochs of private data
server epochs
communication rounds | Adam, lr=0.001, wd=0.0001
256
1
1
1
200 | Adam, lr=0.001, wd=0.0001
256
1
1
1
200 |
| **FedAvg** | optimizer
batch size
client epochs
communication rounds | Adam, lr=0.001, wd=0.0001
64
20
200 | - |
| **FedMD** | optimizer
batch size
client epochs of public data
client epochs of private data
communication rounds | Adam, lr=0.001, wd=0.0001
64
1
2
200 | Adam, lr=0.001, wd=0.0001
64
1
2
200 |
| **FedDF** | optimizer
batch size
client epochs of private data
server epochs
communication rounds | Adam, lr=0.001, wd=0.0001
64
40
5
100 | - |
| **FedGKT** | optimizer
batch size
client epochs of public data
server epochs
communication rounds | Adam, lr=0.001, wd=0.0001
256
1
20
200 | SGD, lr=0.005, wd=0.0001
256
1
40
200 |
| **Cronus** | optimizer
batch size
client epochs of public data
client epochs of private data
communication rounds | Adam, lr=0.001, wd=0.0001
64
1
1
200 | Adam, lr=0.001, wd=0.0001
64
1
1
200 |
| **DS-FL** | optimizer
batch size
client epochs of public data
client epochs of private data
server epochs
communication rounds | Adam, lr=0.001, wd=0.0001
64
1
1
2
300 | Adam, lr=0.001, wd=0.0001
64
1
1
2
300 |
| **Standalone** | optimizer
batch size
epochs | Adam, lr=0.001, wd=0.0001
256
200 | Adam, lr=0.001, wd=0.0001
256
200 |
| **Centralized** | optimizer
batch size
epochs | Adam, lr=0.001, wd=0.0001
256
200 | Adam, lr=0.001, wd=0.0001
256
200 |

**Table 13:** Hyper-parameters used in Experiments on dataset CIFAR-10, CIFAR-100 and CINIC-10

