# OpenReview forum: "FedGEMS: Federated Learning of Larger Server Models via Selective Knowledge Fusion"
_ICLR.cc/2022/Conference — ICLR 2022 Submitted_

### Official Review · Reviewer_Q1kU · 2021-10-31

**Correctness:** 3
**Technical Novelty And Significance:** 3
**Empirical Novelty And Significance:** 2
**Recommendation:** 5
**Confidence:** 4

**Main Review:**

The problem considered in this paper is interesting and important. The need to accommodate on-device systems constraints (limited memory and computational power) significantly limits the performance of models trained using FL. This paper suggests an interesting approach towards overcoming this issue, and the experimental results suggest that the proposed approach is promising.

However the paper has a number of limitations, which are detailed below. Some are related to the practicality of the proposed approach, but the majority of these are limitations of the experimental setup used for comparisons. Normally I would be willing to overlook some of the concerns, but there are so many in this case that my overall recommendation is to reject the paper.

**Major concerns:**
1. The scheme described in Sec 3.3.2 does not seem compatible with privacy considerations that are important in FL. In particular, in order to partition clients into reliable and unreliable sets as described, the server would need to have the individual logits for each training instance from each client. Such information could be pretty easily used to reverse-engineer the clients' training data (e.g., Geiping et al., NeurIPS 2020).

2. If I understand correctly, in all experiments, the distribution of the public data set is identical to that of the private dataset (if the private datasets were all pooled together). This seems very realistic, as one might expect that clients could have data following some distribution locally which is not reflected at all by data in the public dataset; in particular, I'm thinking of clients with private training samples with inputs $x_i$ that fall outside of the distribution of inputs in the public dataset.

3. The experiments use a very small number of clients (only 8), and each has a relatively large amount of data. No information is provided in the main paper about how much training effort is required to achieve the results in Table 2. (How many global rounds, and how many local epochs per round?) Also, the introduction and setup considered seem to imply that this approach is intended for cross-device FL. If that is truly the case, it would be important to include experiments showing that FedGEMS works with many more clients and less data per client. The current results seem to suggest that it may be useful for cross-silo FL.

4. The performance reported for FedGKT is _much_ lower than what is reported in [He et al., 2020](https://arxiv.org/abs/2007.14513). I realize the models may not be exactly the same, but I wouldn't expect to see such a marked difference. This substantial difference, without any remark from the authors, leads me to question how much faith to put in the experimental results. How much effort went into hyperparameter tuning for each method?

5. Are the attacks considered in Sec 4.3 being newly introduced in this paper or have they been considered in previous literature? If they are, references should be included in the main text of the paper (not just in the appendix). If the attacks are actually new, then significantly more motivation should be included for why these attacks are relevant.

6. Table 3 reports the results of poisoning attacks in the homogeneous setting, but it would be much more interesting and relevant to also include the performance in the heterogeneous setting.

7. Fig 3 shows the communication cost per epoch (per round?). This is only part of the story, though, since the overall communication overhead also depends on the number of rounds required to reach the final accuracy. The paper does not report how many rounds each method was run for, or if it's the same for all methods. Upon further examination in Table 8 of the Appendix, it appears that different methods were run for different numbers of rounds. It is not clear that this leads to a fair comparison.

8. Fig 4 shows that the number of samples selected for clients to report back to the server changes during training. Fig 3 shows a fixed communication cost per iteration. Are those numbers at the beginning or end of training, or averaged over an entire training run? Also, wouldn't one expect those values to change substantially depending on the problem and the public/private datasets?

9. The objectives (2), (5), and (6) all depend on the parameter $\epsilon$ that balances between cross-entropy and distillation losses. What value was used in the experiments? It is impossible to reproduce the results without knowing this. Also, how sensitive is performance to the choice of this parameter?

**Minor / requests for additional information:**
* This paper seems to implicitly focus on classification problems. This ought to be stated explicitly in the introduction, and ideally also in the abstract.
* The title ("Client-side...") of Sec 3.3.2 is confusing, since that section describes operations that are all performed at the server.
* Please add an x-label for Fig 3.
* Do the communication amounts for FedGEMS reported in Fig 3 count the overhead of the server sending the selected indices to each client?

**Summary Of The Paper:**

This paper aims to enable training of higher accuracy models by training a high-capacity model at the server and lower capacity models at the clients. Knowledge is shared via distillation on a weighted average of logits provided by the clients, rather than by averaging model parameters or model differences. The approach assumes the existence of a labeled public dataset. Experiments illustrate the promise of this approach.

**Summary Of The Review:**

The problem considered in this paper is interesting and important. The need to accommodate on-device systems constraints (limited memory and computational power) significantly limits the performance of models trained using FL. This paper suggests an interesting approach towards overcoming this issue, and the experimental results suggest that the proposed approach is promising.

However the paper has a number of limitations. Some are related to the practicality of the proposed approach, but the majority of these are limitations of the experimental setup used for comparisons. Normally I would be willing to overlook some of the concerns, but there are so many in this case that my overall recommendation is to reject the paper.

---

> ### Author Response · Authors · 2021-11-23
> **Respond Major Concerns (1-2) to Reviewer Q1kU**
>
> > **[Q1]** ***The scheme described in Sec 3.3.2 does not seem compatible with privacy considerations that are important in FL. In particular, in order to partition clients into reliable and unreliable sets as described, the server would need to have the individual logits for each training instance from each client. Such information could be pretty easily used to reverse-engineer the clients' training data (e.g., Geiping et al., NeurIPS 2020).***
>
> **[Response]** Thanks for your comments. Different from the settings in [1], we only communicate the individual logits of the public dataset which is available to both the server and client models, not the private dataset. In fact, by only communicating public logits, not private models, our KD-based approaches are less vulnerable to privacy leakages from model inversion attacks [1][2], which reverse-engineer the private data from private model parameters. A detailed analysis of the privacy advantage of a KD-based FL approach over the FedAvg-based FL approach has been conducted before in [3], which shows that communicating public logits is more robust to inference attacks. Similar conclusions are applicable here since we only communicate selective public logits.
>
> Another privacy threat for FL is the membership inference attack. To further validate the ability of our selective framework to defend such attacks, we conduct both active and passive membership inference attacks [3] on the CIFAR-10 dataset and the results are shown below. Higher accuracy indicates that the information of private datasets in clients is easier to infer by the attacks. Our performance is much better than FedAvg and is comparable to Cronus which adopts a specific defense mechanism.
>
> | | | | | |
> |---|:--:|:--:|:--:|:--:|
> |**Type**|**FedAvg**|**Cronus**|**FedGEM**|**FedGEMS**|
> |Passive|90.63|58.79|60.22|59.83|
> |Active|92.71|51.63|62.50|59.38|
> | | | | | |
>
> **Table 1:** The accuracy of passive and active membership inference attacks.
>
> > [1] *Geiping, J., Bauermeister, H., Dröge, H. and Moeller, M., 2020. Inverting Gradients--How easy is it to break privacy in federated learning?. arXiv preprint arXiv:2003.14053.*
> >
> > [2] *Zhu, L. and Han, S., 2020. Deep leakage from gradients. In Federated learning (pp. 17-31). Springer, Cham.*
> >
> > [3] *Chang, H., Shejwalkar, V., Shokri, R. and Houmansadr, A., 2019. Cronus: Robust and heterogeneous collaborative learning with black-box knowledge transfer. arXiv preprint arXiv:1912.11279.*
>
> -----
>
> > **[Q2]** ***If I understand correctly, in all experiments, the distribution of the public data set is identical to that of the private dataset (if the private datasets were all pooled together). This seems very realistic, as one might expect that clients could have data following some distribution locally which is not reflected at all by data in the public dataset; in particular, I'm thinking of clients with private training samples with inputs that fall outside of the distribution of inputs in the public dataset.***
>
> **[Response]** We agree that our experiments performed in this work all assume that the public and private datasets are of the same domain. For example, in some real-world scenarios, certain data (images, videos) are publicly available or collected but acquiring more data is challenging due to newly introduced privacy laws such as GDPR. Whether and how this framework can be extended to the setting where private and public datasets are of different distributions is more challenging and will be studied in our future work.

---

> ### Author Response · Authors · 2021-11-23
> **Respond Major Concerns (3) to Reviewer Q1kU**
>
> > **[Q3]** ***The experiments use a very small number of clients (only 8), and each has a relatively large amount of data. No information is provided in the main paper about how much training effort is required to achieve the results in Table 2. (How many global rounds, and how many local epochs per round?) Also, the introduction and setup considered seem to imply that this approach is intended for cross-device FL. If that is truly the case, it would be important to include experiments showing that FedGEMS works with many more clients and less data per client. The current results seem to suggest that it may be useful for cross-silo FL.***
>
> **[Response]** Thanks for your review comments. To illustrate the effectiveness of our framework under more clients and less local data per client, we further conduct experiments with 16 clients, and the data amount per client is reduced by half. The experimental results in Table 2 show that with an increased number of clients and fewer client data, as well as a larger server model (ResNet-56), the performance gain of our approaches is more significant, especially for the client side. Our framework is general and applicable to both cross-silo and cross-device scenarios where the server has more computational power than clients, in which our focus is to improve positive knowledge transfer between client and server. In our revised version, we provide all the experimental details in this new setting.
>
> As for the related hyper-parameters to achieve the results of different methods, such as global rounds (400) and local epochs per round (1), we reported all the hyper-parameters in Appendix C.7 in the previous version. In our revised version, we add the hyper-parameters about our framework FedGEMS in Sec. 4.1 to make it more clear.
>
> |                     |              |             |               |             |              |             |               |             |
> |:-------------------:|:------------:|:-----------:|:-------------:|:-----------:|:------------:|:-----------:|:-------------:|:-----------:|
> |     **Method**      | **Homo**   |             |               |             |  **Hetero**  |             |               |             |
> |                     | **CIFAR-10** |             | **CIFAR-100** |             | **CIFAR-10** |             | **CIFAR-100** |             |
> |                     |  **Server**  | **Clients** |  **Server**   | **Clients** |  **Server**  | **Clients** |  **Server**   | **Clients** |
> |   **Stand-Alone**   |    84.03     |    41.57    |     65.38     |    26.95    |    84.03     |    28.58    |     65.38     |    18.70    |
> |   **Centralized**   |      \-      |    76.21    |      \-       |    58.61    |      \-      |    83.54    |      \-       |    63.44    |
> | **Centralized-All** |    91.27     |    82.45    |     78.13     |    65.51    |    91.27     |    88.54    |     78.13     |    70.96    |
> |     **FedAvg**      |      \-      |    53.33    |      \-       |    31.47    |      \-      |     \-      |      \-       |     \-      |
> |      **FedMD**      |      \-      |    58.47    |      \-       |    34.77    |      \-      |    51.11    |      \-       |    25.57    |
> |     **Cronus**      |      \-      |    57.73    |      \-       |    34.19    |      \-      |    53.47    |      \-       |    29.71    |
> |      **DS-FL**      |      \-      |    50.78    |      \-       |    25.70    |      \-      |    43.83    |      \-       |    16.69    |
> |      **FedDF**      |      \-      |    56.31    |      \-       |    31.27    |      \-      |     \-      |      \-       |     \-      |
> |     **FedGKT**      |    55.67     |     \-      |     29.89     |     \-      |    45.55     |     \-      |     26.96     |     \-      |
> |     **FedGEM**      |    86.62     |    80.35    |     67.72     |    62.61    |    87.11     |    80.14    |     67.27     |    64.73    |
> |     **FedGEMS**     |  **88.08**   |  **81.86**  |   **69.08**   |  **63.81**  |  **87.97**   |  **84.18**  |   **67.72**   |  **65.93**  |
> |                     |              |             |               |             |              |             |               |             |
>
> **Table 2 (Table 2 in our revision):**  Model performance in homogeneous and heterogeneous settings.

---

> ### Author Response · Authors · 2021-11-23
> **Respond Major Concerns (4-6) to Reviewer Q1kU**
>
> > **[Q4]** ***The performance reported for FedGKT is much lower than what is reported in He et al., 2020. I realize the models may not be exactly the same, but I wouldn't expect to see such a marked difference. This substantial difference, without any remark from the authors, leads me to question how much faith to put in the experimental results. How much effort went into hyperparameter tuning for each method?***
>
> **[Response]** Thanks for your questions.
> First, our experiment results are in accordance with results reported in He et al., 2020.
> It is worth noticing that FedGKT only reports the performance of the server model (ResNet56 / ResNet110).
> For example, in the I.I.D. and CIFAR-10 settings, the performance of the server in FedGKT (92.97) is a little bit higher than FedAvg (92.88).
> In our paper, in the I.I.D. and CIFAR-10 settings, the performance of the server in FedGKT (55.67) is also higher than the FedAvg (53.33).
>
> Secondly, we adopted the same public code (FedML, https://github.com/FedML-AI/FedML) to reproduce its reported results with FedML.
> As the result, the accuracy of the server model is 92.51 which is comparable to the accuracy in its original paper, while the accuracy of the client model is 45.23 which was not reported in the paper. Since FedGKT never reports its performance on clients, to avoid further confusion, we remove the FedGKT's client-side performance in our table.
>
> Last but not least, we also tried a series of hyperparameters to tune FedGKT shown in Table 3.
>
> | | | |
> |:---------------------------------:|:----------:|:----------:|
> |        **Hyperparameter**         | **Server** | **Client** |
> |          **lr = 0.001**           |   55.67    |   44.69    |
> |          **lr = 0.005**           |   50.33    |   49.27    |
> |           **lr = 0.01**           |   54.83    |   49.64    |
> |       **Server epoch = 10**       |   53.94    |   40.15    |
> |       **Server epoch = 20**       |   55.67    |   44.69    |
> |       **Server epoch = 40**       |   50.22    |   36.95    |
> | **Pretrain using public dataset** |   55.22    |   38.48    |
> | | | |
>
> **Table 3:** Hyperparameters used in FedGKT.
>
> -----
>
> > **[Q5]** ***Are the attacks considered in Sec 4.3 being newly introduced in this paper or have they been considered in previous literature? If they are, references should be included in the main text of the paper (not just in the appendix). If the attacks are actually new, then significantly more motivation should be included for why these attacks are relevant.***
>
> **[Response]** Our attacks considered in Sec 4.3 are proposed in previous studies [3] to evaluate the robustness of federated learning algorithms. We have added related references in our revised version.
>
> > [3] Chang, H., Shejwalkar, V., Shokri, R. and Houmansadr, A., 2019. Cronus: Robust and heterogeneous collaborative learning with black-box knowledge transfer. arXiv preprint arXiv:1912.11279.
>
> ----
> > **[Q6]** ***Table 3 reports the results of poisoning attacks in the homogeneous setting, but it would be much more interesting and relevant to also include the performance in the heterogeneous setting.***
>
> **[Response]** Thanks for your suggestions.
> We conduct our poisoning attacks in heterogeneous settings and present the corresponding results in Sec. 4.3 of our revised version.
> The results are also shown in Table 4. FedGEMS performs robustly in this setting as well with the highest overall performance.
>
> |             |              |               |              |              |              |               |     |     |
> |:-----------:|:------------:|:-------------:|:------------:|:------------:|:------------:|:-------------:|:---:|:---:|
> | **Method**  |   **PAF**    |               |   **LIE**    |              |   **OFOM**   |               |     |     |
> |             |    Server    |    Client     |    Server    |    Client    |    Server    |    Client     |     |     |
> |  **FedMD**  |      \-      | 16.13 /-68.44 |      \-      | 16.21/-68.28 |      \-      | 22.53/-55.92  |     |     |
> |  **DS-FL**  |      \-      | 37.94/-13.44  |      \-      | 23.42/-46.57 |      \-      | 23.32/-46.79  |     |     |
> | **Cronus**  |      \-      | 53.71/+00.45  |      \-      | 53.98/+00.94 |      \-      | 53.74 /-00.50 |     |     |
> | **FedGKT**  | 13.82/-69.66 |      \-       | 10.42/-77.12 |      \-      | 14.20/-68.83  |      \-       |     |     |
> | **FedGEM**  | 55.94/-35.78 | 72.78/-09.18  | 74.86/-14.06 | 74.32/-07.26 | 85.22/-02.17 | 74.47/-07.08  |     |     |
> | **FedGEMS** | 87.46/-05.80 | 82.58/-01.90  | 87.95/-00.02 | 83.27/-01.08 | 88.53/+00.64 | 82.76/-01.67  |     |     |
> |             |              |               |              |              |              |               |     |     |
>
> **Table 4 (Table 3 in our paper):** Comparison of model poisoning attacks in heterogeneous setting.

---

> > ### Comment · Reviewer_Q1kU · 2021-12-08
> > **Reviewer response**
> >
> > Thanks for your responses (all of them, not just this one).
> >
> > I'm still confused regarding my Q4. I'm looking at Table 2 in the revised paper, where I see the value 55.67 for IID Server on CIFAR-10 that you mentioned. Why is there such a big gap between this value and the 92.97 which is reported in the FedGKT paper? The experiments from both papers now use 16 clients and have similar models.
> >
> > I'm keeping my score unchanged.

---

> > > ### Author Response · Authors · 2021-12-09
> > > **Further Response**
> > >
> > > Thanks for your reply.
> > >
> > > The value 55.67 in Table 2 is different from 92.97 reported in the FedGKT paper because we split the CIFAR-10 into two partitions: public and private datasets.  We use this setting because other baselines, such as FedMD and Cronus,  all adopt a public dataset to transfer knowledge. Therefore, **with only half the data of CIFAR-10 as our experimental setting**, rather than the whole CIFAR-10 in its original paper, the performance of FedGKT drops from 92.97 to 55.67 which makes sense.

---

> > > > ### Comment · Reviewer_Q1kU · 2021-12-09
> > > > **Thanks for the clarification**
> > > >
> > > > I see, thanks for the clarification. In my experience, dropping half the training data (at least in centralized training) generally doesn't not result in such a dramatic decrease in performance, but it may not be completely unreasonable in federated learning.
> > > >
> > > > Based on this and the other responses, which helped clarify some other points, I'm happy to bump up my score by one.

---

> ### Author Response · Authors · 2021-11-23
> **Respond Major Concerns (7-9) to Reviewer Q1kU**
>
> > **[Q7]** ***Fig 3 shows the communication cost per epoch (per round?). This is only part of the story, though, since the overall communication overhead also depends on the number of rounds required to reach the final accuracy. The paper does not report how many rounds each method was run for, or if it's the same for all methods. Upon further examination in Table 8 of the Appendix , it appears that different methods were run for different numbers of rounds. It is not clear that this leads to a fair comparison.***
>
> **[Response]** Thanks for your suggestions. We further evaluate the communication cost at various accuracy levels for each method and the results are shown in Table 5.
> The epochs in both client and server sides among all methods to achieve their best performances are the same as 1 which further ensures fairness.
> More details on parameter-tuning are provided in Appendix B.9. We add these results and discussions in Sec. 4.4 of our revised paper.
>
> |             |                   |                   |                   |                 |
> |:-----------:|------------------:|------------------:|------------------:|:---------------:|
> | **Method**  | **ComU@40% (MB)** | **ComU@50% (MB)** | **ComU@80% (MB)** | **Top-Acc (%)** |
> |  **FedMD**  |            976.56 |          5,493.16 |                \- |      51.11      |
> | **Cronus**  |            610.35 |          2,471.92 |                \- |      53.47      |
> |  **DS-FL**  |          3,692.63 |                \- |                \- |      43.83      |
> | **FedGKT**  |         31,250.00 |                \- |                \- |      45.55      |
> | **FedGEM**  |             91.55 |            244.14 |          3,261.27 |      80.14      |
> | **FedGEMS** |             23.38 |            190.87 |          1,894.84 |      84.18      |
> |             |                   |                   |                   |                 |
>
> **Table 5 (Table 4 in our paper):** Comparison of communication cost and Top-Accuracy. ComU@x: the cumulative communication cost required to achieve the absolute accuracy of x. Top-Accuracy: the highest testing accuracy among the training process.
>
> ----
>
> > **[Q8]** ***Fig 4 shows that the number of samples selected for clients to report back to the server changes during training. Fig 3 shows a fixed communication cost per iteration. Are those numbers at the beginning or end of training, or averaged over an entire training run? Also, wouldn't one expect those values to change substantially depending on the problem and the public/private datasets?***
>
> **[Response]**
> We are sorry for the ambiguity.
> We calculate the averaged communication cost over an entire training run and we make it clear in our revision version. We also evaluated the communication cost per round with respect to different ratios of public/private datasets. The results are shown in Table 6.
>
>
> | | | | | | |
> |:-----------:|:-------:|:-------:|:-------:|:-------:|:-------:|
> | **Method**  | **1:5** | **2:5** | **3:5** | **4:5** | **5:5** |
> | **FedAvg** /  **FedDF**  | 15955.2 |    15955.2    |   15955.2   | 15955.2    |  15955.2     |
> | **FedGKT**  |  1.6M   |   1.6M   |    1.6M   |   1.6M     |   1.6M    |
> | **FedGEM** / **FedMD** / **Cronus** | 6250.0  | 12500.0 | 18750.0 | 25000.0 | 31250.0 |
> | **FedGEMS** | 3230.8  | 6405.8  | 9575.8  | 12733.6 | 15891.3 |
> |  |  | | | | |
>
> **Table 6 (Figure 4 [b] in our paper):** Communication cost per round.
>
> ----
>
> > **[Q9]** ***The objectives (2), (5), and (6) all depend on the parameter that balances between cross-entropy and distillation losses. What value was used in the experiments? It is impossible to reproduce the results without knowing this. Also, how sensitive is performance to the choice of this parameter?***
>
> **[Response]**
> In all experiments, we set $\epsilon=0.75$.
> We make it more clear in our revision version in Sec. 4.1.
> We further conduct a series of experiments to probe the influence of the different choices of this parameter while keeping other experimental settings the same as our main experiment.
> The results are shown in Table 7.
>
> |      |  |  |   |   |   |    |       |   |
> |:-----------:|:----------:|:----------:|:----------:|:----------:|:----------:|:----------:|:----------:|:----------:|
> | **Method**  | *ϵ* = 0.10 |            | *ϵ* = 0.5  |            | *ϵ* = 0.75 |            | *ϵ* = 0.90 |            |
> |             | **Server** | **Client** | **Server** | **Client** | **Server** | **Client** | **Server** | **Client** |
> | **FedGEMS** |   88.71    |   80.37    |   87.78    |   81.97    |   88.08    |   81.86    |   84.50    |   79.53    |
> |      |  |  |   |   |   |    |       |   |
>
> **Table 7 (Table 9 in our paper)**: Comparison of model performance in different $\epsilon$.

---

### Official Review · Reviewer_Ed7U · 2021-11-02

**Correctness:** 3
**Technical Novelty And Significance:** 2
**Empirical Novelty And Significance:** 3
**Recommendation:** 6
**Confidence:** 4

**Main Review:**

Strengths:
- The authors propose an empirical novel selection technique for knowledge fusion.
- The authors tackle multiple challenges in federated learning with their approach.Their results seem to show improved accuracy for both hetero and homogeneous settings. Since only predicted logits are shared, the communication cost is also low.
- The algorithm is also shown to work well in resisting poisoning attacks.

Weakness:
- As per my understanding, the algorithm at the server end uses a combination of techniques. It could be very useful for future researchers to see the impact of each of them separately. For example, we can analyze the effects on accuracy if the logits are not stored (Algorithm 1, Lines 8-9) or the client selection step is not included and all clients are used instead (Algorithm 1, Lines 11-12)
- It looks like the server model is only trained on the public data. What will happen if the server model overfits and predicts everything or most of the samples in the public data correctly? As per my understanding, it will not use the client's information from the logits at all after that.
- The selective knowledge fusion technique is very empirical. The paper therefore needs to have many more ablation studies. For example, the paper could show the effect of larger and smaller server models and the change in accuracy for different numbers of clients during poisoning attacks.
- In Page 5, Paragraph 1, Line 2-3, it is mentioned that the unreliable clients are removed. I understand the reasoning is to remove bad clients but it seems we may be losing some information here. For complex datasets, we may have a scenario where, for outlier samples, generalized clients’ models are not predicting correctly but overfitted clients’ models are predicting better. In that case the server model might not be able to generalize well.
- The title, although technically okay, gives the impression that the server model is very large. But the server model(Resnet-20) is not exactly a whole lot larger than the largest client models (Resnet-11 to Resnet-17). For reference FedGKT uses ResNet-56. As mentioned before, there needs to be more thorough ablation studies for larger and smaller server models.
- In Table 2, the results for FedGKT look to be different from the ones mentioned in the actual FedGKT paper. The exact changes to the training and inference process is not mentioned.



**Summary Of The Paper:**

The paper proposes an empirical technique to train large server models with  selective knowledge fusion. The paper passes logits predicted on a public dataset accessible to both clients and server to share knowledge and uses a selection scheme to withstand poisoning attacks.

**Summary Of The Review:**

This paper proposes a novel approach to tackle several problems in federated learning using selective knowledge fusion. However, the algorithm proposed is largely empirical and needs more thorough ablation studies to strengthen its core claims.

---

> ### Author Response · Authors · 2021-11-23
> **Respond Weakness (1-2) to Reviewer Ed7U**
>
> > **[Q1]** ***As per my understanding, the algorithm at the server end uses a combination of techniques. It could be very useful for future researchers to see the impact of each of them separately. For example, we can analyze the effects on accuracy if the logits are not stored (Algorithm 1, Lines 8-9) or the client selection step is not included and all clients are used instead (Algorithm 1, Lines 11-12)***
>
> **[Response]**
> Thanks for your suggestions.
> We conduct ablation studies to investigate the contribution of each component in our selective strategy.
> In our ablation studies shown in Table 1, the performance of both server and client models decreases.
> This phenomenon indicates that each component contributes to the overall performance.
> Comprehensively, self-training is the most important component and the most probable reason is that learning from the model itself is the most effective way.
> We add these ablation studies and their discussions in Sec. 5.1 of our revised paper.
>
> | | | |
> |:-------------------------|:-------------------:|:-------------------:|
> | **Method**               | **Server Accuracy** | **Client Accuracy** |
> | FedGEMS                  |      **88.08**      |      **81.86**      |
> | \- Self-Training         |        85.73        |        80.17        |
> | \- Self-Distillation     |        86.12        |        81.30        |
> | \- Ensemble Distillation |        86.55        |        80.01        |
> | | | |
>
> **Table 1 (Table 5 in our paper):** Ablation studies of three components in selective strategy.
>
> ----
> > **[Q2]** ***It looks like the server model is only trained on the public data. What will happen if the server model overfits and predicts everything or most of the samples in the public data correctly? As per my understanding, it will not use the client's information from the logits at all after that.***
>
> **[Response]**
> Thanks for your question.
> First, in our framework, the public data serves as an intermediary to transfer logits between server and client models.
> Secondly, both the server and client models are collaborative learning from scratch so that the server is certain to learn from client models in the beginning phase.
> Thirdly, we think that even if the server model can predict most of the samples in the public data, it can also further learn from client models during collaborative learning.
> To further prove our statement, we further conduct a series of experiments where the server model is firstly pre-trained on the public dataset rather than from scratch.
> The results are shown in Table 2.
>
> | | || | | || | |
> |:-------------------------:|:----------:|:----------:|:---:|:---:|:---:|:---:|:---:|:---:|
> |        **Method**         | **Server** | **Client** |     |     |     |     |     |     |
> |      **Stand-Alone**      |   84.03    |   41.57    |     |     |     |     |     |     |
> |      **Centralized**      |     \-     |   76.21    |     |     |     |     |     |     |
> |   **FedGEMS(Pretrain)**   |   87.99    |   78.93    |     |     |     |     |     |     |
> | **FedGEMS(From scratch)** |   88.08    |   81.86    |     |     |     |     |     |     |
> | | || | | || | |
>
> **Table 2:** Comparison of the fine-tuned and from scratch server model.

---

> > ### Comment · Reviewer_Ed7U · 2021-11-29
> > **Thanks for the response**
> >
> > Thanks the authors for the extra experiments. The new results look promising.

---

> ### Author Response · Authors · 2021-11-23
> **Respond Weakness (3-4) to Reviewer Ed7U**
>
> > **[Q3]** ***The selective knowledge fusion technique is very empirical. The paper therefore needs to have many more ablation studies. For example, the paper could show the effect of larger and smaller server models and the change in accuracy for different numbers of clients during poisoning attacks.***
>
> **[Response]**
> The effect of adopting larger and smaller server models was shown in Figure4(b) and discussed in Sec.5.2 in our submitted manuscript, where we conducted a series of experiments varying server models from ResNet-20 to ResNet-56. Here we further adopt ResNet-110 as our server model and the results are shown in Table 3.
>
> ||||
> |:----------------:|:-------------------:|:-------------------:|
> | **Server model** | **Server accuracy** | **Client accuracy** |
> |  **Resnet-20**   |        85.57        |        80.42        |
> |  **Resnet-38**   |        86.88        |        81.53        |
> |  **Resnet-56**   |        88.08        |        81.86        |
> |  **Resnet-110**  |        89.21        |        83.65        |
> ||||
>
> **Table 3 (Figure 5 [a] in our paper):** Model performances in different server sizes.
>
>  As for the change in accuracy for different numbers of clients during poisoning attacks, we adopt 4, 8, and 16 clients to conduct our comparison experiments and the results are shown in Table 4.
>
> |             |             |             |             |             |             |             |     |     |
> |:-----------:|:-----------:|:-----------:|:-----------:|:-----------:|:-----------:|:-----------:|:---:|:---:|
> | **Method**  |      4      |             |      8      |             |     16      |             |     |     |
> |             | **Server**  | **Client**  | **Server**  | **Client**  | **Server**  | **Client**  |     |     |
> | **FedGEMS** | 86.76/-1.06 | 76.36/-4.29 | 87.07/-0.75 | 82.57/-2.05 | 87.95/-0.02 | 83.27/-0.18 |     |     |
> |             |             |             |             |             |             |             |     |     |
>
> **Table 4:** Comparison of the effects of the different numbers of clients on poisoning attack.
>
> ----
>
> > **[Q4]** ***In Page 5, Paragraph 1, Line 2-3, it is mentioned that the unreliable clients are removed. I understand the reasoning is to remove bad clients but it seems we may be losing some information here. For complex datasets, we may have a scenario where, for outlier samples, generalized clients’ models are not predicting correctly but overfitted clients’ models are predicting better. In that case the server model might not be able to generalize well.***
>
> **[Response]** Thanks for your questions.
> First, in our previous version, we have compared FedGEM with FedGEMS in Table 2 to show that the selective knowledge fusion module brings an improvement to server model.
> The former model absorbs the knowledge from all the clients but damages the overall performance, which means that negative influence (poison) is more prominent than positive influence (generalization).
> Secondly, we further conduct an ablation study by removing the ensemble distillation components to validate the statement.
> The results are shown in Table 5.
>
> ||||
> |:-------------------------|:-------------------:|:-------------------:|
> | **Method**               | **Server Accuracy** | **Client Accuracy** |
> | FedGEMS                  |      **88.08**      |      **81.86**      |
> | \- Ensemble Distillation |        86.55        |        80.01        |
> ||||
>
> **Table 5:** Ablation study of ensemble distillation component.

---

> ### Author Response · Authors · 2021-11-23
> **Respond Weakness (5) to Reviewer Ed7U**
>
> > **[Q5]** ***The title, although technically okay, gives the impression that the server model is very large. But the server model(Resnet-20) is not exactly a whole lot larger than the largest client models (Resnet-11 to Resnet-17). For reference FedGKT uses ResNet-56. As mentioned before, there needs to be more thorough ablation studies for larger and smaller server models.***
>
> **[Response]** Thanks for your suggestions. To avoid confusion, we have slightly changed our introduction in our revision to explain the larger server model (GEM) is used to denote that the server model is larger than the client model. In our original manuscript, we conducted a series of experiments varying the server model from ResNet-20 to ResNet-56 in Sec. 5.2 and Figure 4(b). In our revision, we include the experimental evaluations with a larger server model ResNet-56, and more number of clients. The results are shown in Table 6. We further evaluate the performance of our framework using ResNet-110, shown in Table 3 under [Q3].
>
> |                     |              |             |               |             |              |             |               |             |
> |:-------------------:|:------------:|:-----------:|:-------------:|:-----------:|:------------:|:-----------:|:-------------:|:-----------:|
> |     **Method**      | **Homo**   |             |               |             |  **Hetero**  |             |               |             |
> |                     | **CIFAR-10** |             | **CIFAR-100** |             | **CIFAR-10** |             | **CIFAR-100** |             |
> |                     |  **Server**  | **Clients** |  **Server**   | **Clients** |  **Server**  | **Clients** |  **Server**   | **Clients** |
> |   **Stand-Alone**   |    84.03     |    41.57    |     65.38     |    26.95    |    84.03     |    28.58    |     65.38     |    18.70    |
> |   **Centralized**   |      \-      |    76.21    |      \-       |    58.61    |      \-      |    83.54    |      \-       |    63.44    |
> | **Centralized-All** |    91.27     |    82.45    |     78.13     |    65.51    |    91.27     |    88.54    |     78.13     |    70.96    |
> |     **FedAvg**      |      \-      |    53.33    |      \-       |    31.47    |      \-      |     \-      |      \-       |     \-      |
> |      **FedMD**      |      \-      |    58.47    |      \-       |    34.77    |      \-      |    51.11    |      \-       |    25.57    |
> |     **Cronus**      |      \-      |    57.73    |      \-       |    34.19    |      \-      |    53.47    |      \-       |    29.71    |
> |      **DS-FL**      |      \-      |    50.78    |      \-       |    25.70    |      \-      |    43.83    |      \-       |    16.69    |
> |      **FedDF**      |      \-      |    56.31    |      \-       |    31.27    |      \-      |     \-      |      \-       |     \-      |
> |     **FedGKT**      |    55.67     |     \-      |     29.89     |     \-      |    45.55     |     \-      |     26.96     |     \-      |
> |     **FedGEM**      |    86.62     |    80.35    |     67.72     |    62.61    |    87.11     |    80.14    |     67.27     |    64.73    |
> |     **FedGEMS**     |  **88.08**   |  **81.86**  |   **69.08**   |  **63.81**  |  **87.97**   |  **84.18**  |   **67.72**   |  **65.93**  |
> |                     |              |             |               |             |              |             |               |             |
>
> **Table 6 (Table 2 in our paper):**  Model performance in homogeneous and heterogeneous settings.

---

> ### Author Response · Authors · 2021-11-23
> **Respond Weakness (6) to Reviewer Ed7U**
>
> > **[Q6]** ***In Table 2, the results for FedGKT look to be different from the ones mentioned in the actual FedGKT paper. The exact changes to the training and inference process is not mentioned.***
>
> **[Response]**
> Thanks for your questions.
> First, our experiment results are in accordance with results reported in He et al., 2020.
> It is worth noticing that FedGKT only reports the performance of the server model (ResNet56 / ResNet110).
> For example, in the I.I.D. and CIFAR-10 settings, the performance of the server in FedGKT (92.97) is a little bit higher than FedAvg (92.88).
> In our paper, in the I.I.D. and CIFAR-10 settings, the performance of the server in FedGKT (55.67) is also higher than the FedAvg (53.33).
>
> Secondly, we adopted the same public code (FedML, https://github.com/FedML-AI/FedML) to reproduce its reported results with FedML.
> As the result, the accuracy of the server model is 92.51 which is comparable to the accuracy in its original paper, while the accuracy of the client model is 45.23 which was not reported in the paper. Since FedGKT never reports its performance on clients, to avoid further confusion, we remove the FedGKT's client-side performance in our table.
>
> Last but not least, we also tried a series of hyperparameters to tune FedGKT shown in Table 7.
>
> | | | |
> |:---------------------------------:|:----------:|:----------:|
> |        **Hyperparameter**         | **Server** | **Client** |
> |          **lr = 0.001**           |   55.67    |   44.69    |
> |          **lr = 0.005**           |   50.33    |   49.27    |
> |           **lr = 0.01**           |   54.83    |   49.64    |
> |       **Server epoch = 10**       |   53.94    |   40.15    |
> |       **Server epoch = 20**       |   55.67    |   44.69    |
> |       **Server epoch = 40**       |   50.22    |   36.95    |
> | **Pretrain using public dataset** |   55.22    |   38.48    |
> | | | |
>
> **Table 7:** Hyperparameters used in FedGKT.

---

> ### Comment · Reviewer_Ed7U · 2021-11-29
> **Thanks the authors for the response**
>
> Thanks the authors for the response. I have read the authors' response as well as the comments from other reviewers. The authors have addressed most of my concerns. I would like to raise my score for 1 more point. But I do have a concern about how largely the performance of the proposed method is dependent on the size of the public dataset.

---

> > ### Author Response · Authors · 2021-11-30
> > **Thanks Letter for Review**
> >
> > Dear Reviewer,
> >
> > Thanks for your valuable advice. We appreciate your suggestions which help us refine our work and are encouraged by your positive feedback.
> >
> > As for the performance in different sizes of public datasets, we discussed the effect of public dataset size in Sec. 5.3 in our manuscript, see Figure 5. We re-implement this experiment under our new experiment setting with 16 clients and ResNet-56 as server model, and the results are shown in the following Table 8.
> >
> > Overall, compared with all the KD-based methods (FedDF, FedMD, Cronus, and DS-FL) using the same size of public dataset, our proposed framework FedGEMS consistently surpasses them by a large margin with different ratios. This phenomenon proves that our proposed method FedGEMS is efficient under the same situation.
> >
> > However, as you point out, the size of public dataset has an impact on our performance, so we will further explore methodologies to reduce the dependence on public dataset in our future work.
> >
> > Best Regards.
> >
> > |             |            |            |            |            |            |            |            |            |            |            |     |     |     |     |
> > |:-----------:|:----------:|:----------:|:----------:|:----------:|:----------:|:----------:|:----------:|:----------:|:----------:|:----------:|:---:|:---:|:---:|:---:|
> > | **Method**  |  **5000**  |            | **10000**  |            | **15000**  |            | **20000**  |            | **25000**  |            |     |     |     |     |
> > |             | **Server** | **Client** | **Server** | **Client** | **Server** | **Client** | **Server** | **Client** | **Server** | **Client** |     |     |     |     |
> > |  **FedDF**  |   38.98    |   39.73    |   49.14    |   47.14    |   49.00    |   47.59    |   49.28    |   48.00    |   56.41    |   56.31    |     |     |     |     |
> > |  **FedMD**  |     \-     |   48.74    |     \-     |   52.02    |     \-     |   54.32    |     \-     |   56.00    |     \-     |   58.47    |     |     |     |     |
> > | **FedGEM**  |   65.54    |   63.38    |   76.71    |   70.76    |   81.68    |   71.08    |   85.15    |   71.28    |   86.62    |   80.35    |     |     |     |     |
> > | **FedGEMS** |   65.57    |   63.16    |   78.45    |   74.59    |   81.93    |   77.48    |   85.49    |   80.47    |   88.08    |   81.86    |     |     |     |     |
> > |             |            |            |            |            |            |            |            |            |            |            |     |     |     |     |
> >
> > **Table 8:** Model performance in homogeneous setting with different public dataset sizes.

---

### Official Review · Reviewer_Cj8x · 2021-11-06

**Correctness:** 3
**Technical Novelty And Significance:** 3
**Empirical Novelty And Significance:** 3
**Recommendation:** 6
**Confidence:** 3

**Main Review:**

* Strengths
1. The selective KD is novel to FL
2. The problem (training a large-scale model) by utilizing scatter data from large number of edge devices is well motivated.
3. Many important and highly related baselines are compared.

* Weaknesses
1. It's supervising to me that a smaller model (ResNet-20) used in the proposed framework can outperform a larger model (ResNet-56) used in  FedDF and FedGKT.

2. FedGKT's accuracy is not the same as the result reported in the original paper. Is that because of bug or not well-tuned?

3. The proposed framework is not data-free KD like FedGKT. The public and private data is 1:1. In practice, this may not be the case. Another issue of such setting is that the performance in other ratio is not clear. It would be better to do an ablation study to understand the performance in different ratio.

4. The communication cost is over claimed. FedGEMS requires a public dataset for each client to boost the transfer, when the number of clients is big, the cost is very high. The total communication cost should be larger than FedGKT. The benefit of FedGKT is that it only replies on the sample number on each client, which is normally small. Another concern is the memory cost. FedGKT does not need any public data to do KD, however FedGEMS requires too many samples for edge devices (30000 in the current implementation).

5. Another issue would be the scalability. Note that cross-device FL should be stateless, meaning that the newly sampled clients do not have any previous cached states (optimizer states, logiits, etc) [1] . FedGEMS's assumption seems can only be used in cross-silo FL.

6. It seems the author uses FedML to finish the experiments. Does FedML support the implementation? Especially the complex communication and KD pattern described in FedGEMS.

7. It's highly appreciated when the authors can discuss some limitations. For example, the memory cost for public data; whether the client side can support a large model; what if the public and private data ratio is changed, etc.

8. The Introduction attempt to convince readers that training GPT-3 like giant model in FL is necessary. This is ok to me. But the experiments only talk about small scale ResNet-20. Note that GPT-3 is a billion-level model, while ResNet-20 is much smaller. This somehow contradicts the motivation. Another related issue is that the paper is titled as "larger" model but the model used in experiments (ResNet-20) is smaller than other baselines.

9. The writing need to be further polished.

10. "Continual" in Abstract may not be a correct word, since it normally refers to continual ML. But FedGEMS does not have related design.

[1] Federated Reconstruction: Partially Local Federated Learning. NeurIPS 2021.


**Summary Of The Paper:**

This work proposed a KD-based FL framework for training a larger server model securely. The proposed framework is a combination of FedGKT-like idea and further apply some KD strategies for selective transfer. Many baselines are compared to demonstrate its efficacy.

**Summary Of The Review:**

Overall, it's a good empirical paper, but it's better to address some of the above concerns before applying the proposed framework in practice.

---

> ### Author Response · Authors · 2021-11-23
> **Respond Weakness (1-2) to Reviewer Cj8x**
>
> > **[Q1]** ***It's supervising to me that a smaller model (ResNet-20) used in the proposed framework can outperform a larger model (ResNet-56) used in FedDF and FedGKT.***
>
> **[Response]** We are sorry for the confusion.
> In our manuscript, we stated that ``the server models in FedDF and FedGKT are ResNet-11 (same as client models) and ResNet-56 respectively, following their own designs'' in Section 4.1.
> Therefore, the server model in FedDF is ResNet-11 and our performance of both server and client models are higher than FedDF.
> As for FedGKT, the server model is ResNet-56 and we will elaborately explain the phenomena in [R3-O2].
>
> ----
> > **[Q2]** ***FedGKT's accuracy is not the same as the result reported in the original paper. Is that because of bug or not well-tuned?***
>
> **[Response]**
> Thanks for your questions. First, our experiment results are in accordance with results reported in He et al., 2020. It is worth noticing that FedGKT only reports the performance of the server model (ResNet56 / ResNet110). For example, in the I.I.D. and CIFAR-10 settings, the performance of the server in FedGKT (92.97) is a little bit higher than FedAvg (92.88). In our paper, in the I.I.D. and CIFAR-10 settings, the performance of the server in FedGKT (55.67) is also higher than the FedAvg (53.33).
>
> Secondly, we adopted the same public code (FedML, https://github.com/FedML-AI/FedML) to reproduce its reported results with FedML. As the result, the accuracy of the server model is 92.51 which is comparable to the accuracy in its original paper, while the accuracy of the client model is 45.23 which was not reported in the paper. Since FedGKT never reports its performance on clients, to avoid further confusion, we remove the FedGKT's client-side performance in our tables.
>
> Last but not least, we also tried a series of hyperparameters to tune FedGKT shown in Table 1.
>
> | | | |
> |:---------------------------------:|:----------:|:----------:|
> |        **Hyperparameter**         | **Server** | **Client** |
> |          **lr = 0.001**           |   55.67    |   44.69    |
> |          **lr = 0.005**           |   50.33    |   49.27    |
> |           **lr = 0.01**           |   54.83    |   49.64    |
> |       **Server epoch = 10**       |   53.94    |   40.15    |
> |       **Server epoch = 20**       |   55.67    |   44.69    |
> |       **Server epoch = 40**       |   50.22    |   36.95    |
> | **Pretrain using public dataset** |   55.22    |   38.48    |
> | | | |
>
> **Table 1:** Hyperparameters used in FedGKT.

---

> > ### Comment · Reviewer_Cj8x · 2021-11-24
> > **about FedGKT**
> >
> > It seems you misunderstand FedGKT's design goal. There is no need to discuss client-model partition's performance. Its goal is to get a large model but train on resource-constrained devices, meaning that the final output is a large model that can be deployed to the client by remote inference or on-device inference. This is possible because inference resource cost is much smaller than the training (due to activation maps, optimizer states, gradient). As such, only discussing its entire model performance is enough.
> >
> > Following this angle, I am confused why both client and server model should be evaluated?

---

> > > ### Author Response · Authors · 2021-11-30
> > > **about FedGKT**
> > >
> > > > ***[C1] It seems you misunderstand FedGKT's design goal. There is no need to discuss client-model partition's performance. Its goal is to get a large model but train on resource-constrained devices, meaning that the final output is a large model that can be deployed to the client by remote inference or on-device inference. This is possible because inference resource cost is much smaller than the training (due to activation maps, optimizer states, gradient). As such, only discussing its entire model performance is enough.
> > > Following this angle, I am confused why both client and server model should be evaluated?***
> > >
> > > **[Response]** Thanks for your suggestions. Considering that our proposed framework FedGEMS' design goal is to adopt a larger server model to boost the performance of both server and client models, we report the performance of both server and client models for our baselines including FedGKT and FedDF, to ensure consistency under our experiment settings. However, that might not exactly reflect FedGKT's design purpose and introduce some confusion to readers. Therefore, we fix the problems in our revised version. We thank your kind reminder and meaningful advice.

---

> ### Author Response · Authors · 2021-11-23
> **Respond Weakness (3) to Reviewer Cj8x**
>
> ----
> > **[Q3]** ***The proposed framework is not data-free KD like FedGKT. The public and private data is 1:1. In practice, this may not be the case. Another issue of such setting is that the performance in other ratio is not clear. It would be better to do an ablation study to understand the performance in different ratio.***
>
> **[Response]**
> Thanks for your advice. As for the performance in other ratios, we discussed the effect of public dataset size in Sec. 5.3 in our manuscript, see Figure 5. Here the ratio of the public dataset to the private dataset is changing while the public dataset size is increasing. In our revision, we change our metric from public data size to "the ratio of public dataset to private dataset'' to make it more clear.
> The results of different ratios are shown in Table 3.
>
> Most existing KD-based methods in federated learning, including FedMD, FedDF, etc, still adopt public datasets to transfer knowledge.
> Data-free is the main advantage of FedGKT, however, the extra extractors transmitted between the clients and the server introduce new privacy concerns as mentioned in FedGKT's original paper, and the performance of FedGKT is superior to our approach as shown in our results. We will further improve our method to reduce the dependence on the public dataset in our future work.
>
> |             |            |            |            |            |            |            |            |            |            |            |     |     |     |     |
> |:-----------:|:----------:|:----------:|:----------:|:----------:|:----------:|:----------:|:----------:|:----------:|:----------:|:----------:|:---:|:---:|:---:|:---:|
> | **Method**  |  **1:5**   |            |  **2:5**   |            |  **3:5**   |            |  **4:5**   |            |  **5:5**   |            |     |     |     |     |
> |             | **Server** | **Client** | **Server** | **Client** | **Server** | **Client** | **Server** | **Client** | **Server** | **Client** |     |     |     |     |
> |  **FedDF**  |   38.98    |   39.73    |   49.14    |   47.14    |   49.00    |   47.59    |   49.28    |   48.00    |   56.41    |   56.31    |     |     |     |     |
> |  **FedMD**  |     \-     |   48.74    |     \-     |   52.02    |     \-     |   54.32    |     \-     |   56.00    |     \-     |   58.47    |     |     |     |     |
> | **FedGEM**  |   65.54    |   63.38    |   76.71    |   70.76    |   81.68    |   71.08    |   85.15    |   71.28    |   86.62    |   80.35    |     |     |     |     |
> | **FedGEMS** |   65.57    |   63.16    |   78.45    |   74.59    |   81.93    |   77.48    |   85.49    |   80.47    |   88.08    |   81.86    |     |     |     |     |
> |             |            |            |            |            |            |            |            |            |            |            |     |     |     |     |
>
> **Table 2 (Figure 4 [a] in our paper):** Model performance in homogeneous setting with different public and private data ratios.

---

> > ### Comment · Reviewer_Cj8x · 2021-11-24
> > **ratio is still the concern**
> >
> > Thanks for showing the results of different ratios. However, I can see the accuracy is very limited (lower than using LeNet which can achieve more than 80% in CIFAR-10) when the size of public data is relatively small. As for the motivation of FL, we don't have enough public data to train a good model, and then we need to do FL. Therefore, requiring a large ratio of a public dataset does not match the practice.
> >
> > Overall, I like your evaluation very much. It clearly shows the limitation of the proposed framework.

---

> > > ### Author Response · Authors · 2021-11-30
> > > **Further response**
> > >
> > > > ***[C1] Thanks for showing the results of different ratios. However, I can see the accuracy is very limited (lower than using LeNet which can achieve more than 80% in CIFAR-10) when the size of public data is relatively small. As for the motivation of FL, we don't have enough public data to train a good model, and then we need to do FL. Therefore, requiring a large ratio of a public dataset does not match the practice.
> > > Overall, I like your evaluation very much. It clearly shows the limitation of the proposed framework.***
> > >
> > > **[Response]** Thanks for your advice.
> > >
> > > We would like to point out that LeNet achieves more than 80\% via the centralized training using the whole CIFAR-10 dataset, while our FedGEMS split CIFAR-10 into public and private datasets and allocate the private dataset to each client. Therefore, it is reasonable that our performance using a small public data size is not reaching the performance using the entire dataset.
> > >
> > > Compared with all the KD-based methods (FedDF, FedMD, Cronus, and DS-FL) using the same size of public dataset, our proposed framework FedGEMS consistently surpasses them by a large margin with different ratios. This phenomenon proves that our proposed method FedGEMS is efficient in the same situation.
> > >
> > > However, the size of public dataset has an impact on our performance, we will further improve our method to reduce the dependence on public dataset in our future work.

---

> ### Author Response · Authors · 2021-11-23
> **Respond Weakness (4-6) to Reviewer Cj8x**
>
> > **[Q4]** ***The communication cost is over claimed. FedGEMS requires a public dataset for each client to boost the transfer, when the number of clients is big, the cost is very high. The total communication cost should be larger than FedGKT. The benefit of FedGKT is that it only replies on the sample number on each client, which is normally small. Another concern is the memory cost. FedGKT does not need any public data to do KD, however FedGEMS requires too many samples for edge devices (30000 in the current implementation).***
>
> **[Response]** Although FedGKT does not require a public dataset to transfer knowledge, it needs to transfer both feature maps and logits of the private dataset from clients to the server. Specifically, the communication cost per feature map is 64kb, while the communication cost per logits is only 0.039kb. Due to the expensive communication cost for feature maps, the overall communication cost per round of FedGKT is 1.6M (kb), which is the highest among all the methods. Benefiting from our selective strategy, we only averagely adopt 426 samples in the whole 25000 samples of the public dataset to transfer knowledge and our communication cost per round is 15891.1kb.
> We also evaluated the communication cost per round with respect to different ratios of public/private datasets. The results are shown in Table 3.
>
> | | | | | | |
> |:-----------:|:-------:|:-------:|:-------:|:-------:|:-------:|
> | **Method**  | **1:5** | **2:5** | **3:5** | **4:5** | **5:5** |
> | **FedAvg** /  **FedDF**  | 15955.2 |    15955.2    |   15955.2   | 15955.2    |  15955.2     |
> | **FedGKT**  |  1.6M   |   1.6M   |    1.6M   |   1.6M     |   1.6M    |
> | **FedGEM** / **FedMD** / **Cronus** | 6250.0  | 12500.0 | 18750.0 | 25000.0 | 31250.0 |
> | **FedGEMS** | 3230.8  | 6405.8  | 9575.8  | 12733.6 | 15891.3 |
> |  |  | | | | |
>
> **Table 3 (Figure 4 [b] in our paper):** Communication cost per round with different ratios.
>
> As for memory cost, we report three related metrics of both FedGKT and our FedGEMS recording in wandb (a central dashboard, https://wandb.ai/site).
> The averaged memory cost are shown in Table 4.
> All the memory costs of our proposed framework FedGEMS are lower than FedGKT and the most probably extra cost is caused by the extractors and their feature maps.
>
> | | | | | |||||
> |:-----------:|:--------------------------------:|:------------------------:|:-----------------------------:|:---:|:---:|:---:|:---:|:---:|
> | **Method**  | **Process GPU memory allocated** | **GPU memory allocated** | **System memory utilization** |     |     |     |     |     |
> | **FedGKT**  |              37.81%              |          37.81%          |            79.82%             |     |     |     |     |     |
> | **FedGEMS** |              29.28%              |          29.28%          |            70.90%             |     |     |     |     |     |
> | | | | | |||||
>
> **Table 4:** Memory cost of FedGKT and FedGEMS.
>
> ----
>
> > **[Q5]** ***Another issue would be the scalability. Note that cross-device FL should be stateless, meaning that the newly sampled clients do not have any previous cached states (optimizer states, logiits, etc) [1] . FedGEMS's assumption seems can only be used in cross-silo FL.***
>
> **[Response]** Thanks for your comments.
> In our proposed framework FedGEMS, we do not need to save any previous cached states on the client side, but only the cached previous logits on the server side to execute self-distillation. That is, clients are newly sampled in each round and stateless. Therefore, our FedGEMS is suitable for both cross-device and cross-silo FL.
>
> ----
> > **[Q6]** ***It seems the author uses FedML to finish the experiments. Does FedML support the implementation? Especially the complex communication and KD pattern described in FedGEMS.***
>
> **[Response]** Thanks for your questions.
> We implemented our frameworks and other KD-based FL baselines based on FedML. We made necessary modifications to FedML for implementing the selective module and the knowledge fusion. Our code will be publicly available soon.

---

> > ### Comment · Reviewer_Cj8x · 2021-11-24
> > **Further response**
> >
> > I am ok with the communication analysis. But I don't think comparing the overall communication usage in unit of bytes is enough to convey the system-wise benefit. When the goal is the trade-off between communication and memory constraints/computational speed, the communication cost (in units) is not the main concern, especially in cross-silo setting where the bandwidth is high (no communication charge whatever cost is). Even in cross-device setting, the main bottleneck of a slow training is due to the local computational time (no GPU accelerator). Normally, in such a setting, the communication is not the domination, given that now we have 5G/6G high bandwidth communication. So, achieving a high performant model using a smaller client model to accelerate the training and avoid the edge memory limitation is the goal in practice. Please consider communication frequency and payload size, not just the total communication cost. Frequent communication and larger payload size may also lead to serious straggler problem. For example, FedGKT uses much smaller payload size and low frequent communication. As such, I suggest authors adding experiments for an end-to-end training time comparison under similar hyperparameters.
> >
> > Any experiments to support Q5? For cross-device, it's better to train on different cohort sizes at numerous clients.

---

> > > ### Author Response · Authors · 2021-11-30
> > > **Further response**
> > >
> > > > ***[C1] I am ok with the communication analysis. But I don't think comparing the overall communication usage in unit of bytes is enough to convey the system-wise benefit. When the goal is the trade-off between communication and memory constraints/computational speed, the communication cost (in units) is not the main concern, especially in cross-silo setting where the bandwidth is high (no communication charge whatever cost is). Even in cross-device setting, the main bottleneck of a slow training is due to the local computational time (no GPU accelerator). Normally, in such a setting, the communication is not the domination, given that now we have 5G/6G high bandwidth communication. So, achieving a high performant model using a smaller client model to accelerate the training and avoid the edge memory limitation is the goal in practice. Please consider communication frequency and payload size, not just the total communication cost. Frequent communication and larger payload size may also lead to serious straggler problem. For example, FedGKT uses much smaller payload size and low frequent communication. As such, I suggest authors adding experiments for an end-to-end training time comparison under similar hyperparameters.***
> > >
> > > **[Response]** Thanks for your suggestions. We conduct additional experiments to compare the training time at various accuracy levels under the same hyperparameters as our communication analysis. The results are shown in Table 5. As for 40\% accuracy, the training time of our proposed framework FedGEMS is much shorter than all other baselines including FedGKT, which indicates that the negative influence of communication frequency from synchronous setting in FedGEMS can be negligible during the training process in our simulation with 8 NVIDIA Tesla V100-SXM2 GPUs. We acknowledge that the straggle problem is an important topic that can not be fully covered in this work. How it will affect our framework in real-world cross-device settings with a large number of edge devices deserves further studies.
> > >
> > > | | | | | |
> > > |:------------------------:|-----------------:|-----------------:|-----------------:|:---------------:|
> > > |        **Method**        | **Time@40% (s)** | **Time@50% (s)** | **Time@80% (s)** | **Top-Acc (%)** |
> > > |        **FedMD**         |           497.76 |         1,431.06 |               \- |      51.11      |
> > > |        **Cronus**        |         1,368.00 |         5,540.40 |               \- |      53.47      |
> > > |        **DS-FL**         |        17,327.40 |               \- |               \- |      43.83      |
> > > |        **FedGKT**        |         3,179.90 |               \- |               \- |      45.55      |
> > > |        **FedGEM**        |           370.20 |         1,234.00 |        43,190.00 |      80.14      |
> > > | **FedGEMS (16 Clients)** |           365.15 |           821.59 |        10,498.12 |      84.18      |
> > > | **FedGEMS (32 Clients)** |           509.00 |         1,145.25 |        33,339.50 |      81.10      |
> > > | **FedGEMS (64 Clients)** |         2,018.75 |         4,441.25 |        96,900.00 |      81.62      |
> > > | | | | | |
> > >
> > > **Table 5:** Comparison of training time. Time@x: the cumulative training time required to achieve the absolute accuracy of x. Top-Accuracy: the highest testing accuracy among the training process.
> > >
> > > ----
> > >
> > > > ***[C2] Any experiments to support Q5? For cross-device, it's better to train on different cohort sizes at numerous clients.***
> > >
> > > **[Response]** Thanks for your suggestions. We have already conducted a series of experiments with different client numbers and updated the analysis in Sec. 5.4 (effect of the number of clients) of our revised version.
> > > We vary the number of clients from 4 to 64 of our proposed method FedGEMS while keeping the total number of private samples the same. Thus the number of private samples per client varies accordingly.
> > > The experimental results indicate that with increased number of clients and less client data, the performance of our approaches is basically stable especially on client side, benefiting from our larger server model setting with fused knowledge.
> > > The results are shown in the Table 6.
> > >
> > > |||||||
> > > |:----------:|:-----:|:-----:|:-----:|:-----:|:-----:|
> > > |            |   4   |   8   |  16   |  32   |  64   |
> > > | **Server** | 87.07 | 87.75 | 88.08 | 86.74 | 87.81 |
> > > | **Client** | 81.91 | 81.72 | 81.86 | 81.10 | 81.62 |
> > > |||||||
> > >
> > > **Table 6 (Figure 5[b] in our paper):** Comparison of the different number of clients of FedGEMS.

---

> ### Author Response · Authors · 2021-11-23
> **Respond Weakness (7-10) to Reviewer Cj8x**
>
> > **[Q7]** ***It's highly appreciated when the authors can discuss some limitations. For example, the memory cost for public data; whether the client side can support a large model; what if the public and private data ratio is changed, etc.***
>
> **[Response]** Thanks for your suggestions.
> We discuss the memory cost under [Q4] in detail.
> As for large models on client side, our motivation is to break through the model capacity of FL due to the constrained computation resources on client side. Therefore, we assume that clients are typically edge devices such as mobile or IoT devices, which can not have a large model.
> As for the public and private data ratio, we discussed the effect of public dataset size in Sec. 5.2 and Figure 4 in our revised version, and we further add more experimental details and results, shown also under [Q3].
>
> One of the limitations of our approach is the usage of the labeled public dataset. Another limitation is that the public and private datasets are from the same domain. It would be interesting to further explore using only unlabeled or few public datasets, as well as using public data from a different domain. We have added this discussion of limitations in our conclusion.
>
> ----
> > **[Q8]** ***The Introduction attempt to convince readers that training GPT-3 like giant model in FL is necessary. This is ok to me. But the experiments only talk about small scale ResNet-20. Note that GPT-3 is a billion-level model, while ResNet-20 is much smaller. This somehow contradicts the motivation. Another related issue is that the paper is titled as "larger" model but the model used in experiments (ResNet-20) is smaller than other baselines.***
>
> **[Response]** Thanks for your comments. First of all, we clarify that the word ``larger'' in our title and introduction is used to indicate that our server model is larger than client models, as shown in Table 1 in our manuscript. However, we agree that ResNet-20 is somehow small. To reduce confusion and enhance our experiments,  we explored ResNet-56 and ResNet-110 as our server model in our revision. We re-implement all our experiments and update all our results accordingly.
>
> ----
> > **[Q9]** ***The writing need to be further polished.***
>
> **[Response]** Thank you for your suggestions.
> We have carefully polished the paper to make it easier to understand.
>
> ----
> > **[Q10]** ***"Continual" in Abstract may not be a correct word, since it normally refers to continual ML. But FedGEMS does not have related design.***
>
> **[Response]** Thanks for your advice.
> We refine our expressions to avoid ambiguity in our revision version.

---

> > ### Comment · Reviewer_Cj8x · 2021-11-24
> > **Further Response**
> >
> > Dear Authors,
> >
> > I am satisfied with your answers in this part.
> >
> > Most of my confusion has been cleared, thus I decide to raise 1 score. I can see the paper has its benefit in practice, though the scope is limited.

---

> > > ### Author Response · Authors · 2021-11-30
> > > **Thanks Letter for Review**
> > >
> > > Dear Reviewer,
> > >
> > > Thanks for your response.
> > > We really appreciate and are encouraged by your decision. Your valuable suggestions help us refine our work.
> > > We will devote ourselves to solving the remaining limitations in our future work.
> > >
> > > Best Regards.

---

### Official Review · Reviewer_tN4o · 2021-11-09

**Correctness:** 2
**Technical Novelty And Significance:** 2
**Empirical Novelty And Significance:** 2
**Recommendation:** 3
**Confidence:** 4

**Main Review:**

++ Many experiments are conducted to show the results on performance, robustness against poisoning attacks, comm. cost, the model behavior in different server sizes.

However, major concerns are existing in the current form of this paper:

-- The contribution is not clear. The key issues of existing methods and the challenges of the proposed method are not clearly stated.

-- The technical depth seems weak. Table 1 is a good summary, but it also shows the weakness of the proposed model.

Smaller ones:

The mathematical expressions need to be improved, which hinders the understanding of the content.

The notations are not consistent. For example, f_s and f^0 are both used to denote the server model.

No explanations for symbols L_DL, \epsilon, et al.

The y_i in Section 3.1 should be y_i^k


**Summary Of The Paper:**

This paper tries to address the big model on the server-side in federated learning (FL) via knowledge distillation. Selective transfer knowledge happens on both server and clients sides. The proposed model maintains a set of global, correct logits so that it can be used as memories for the server's distillation. When the server predicts wrong on the public datasets, it transfers the knowledge from the correct, weighted clients. The proposed model is evaluated on image datasets with many FL baselines.

**Summary Of The Review:**

contribution not clear, technical depth weak, writing and notation to be improved.

---

> ### Author Response · Authors · 2021-11-23
> **Respond Major Concerns (1-2) to Reviewer tN4o**
>
> > **[Q1]** ***The contribution is not clear. The key issues of existing methods and the challenges of the proposed method are not clearly stated.***
>
> **[Response]** Thanks for your comments. The contribution is stated in Introduction Paragraph \#4, the key issues of existing methods and challenges are stated in Introduction Paragraph \#2 and \#3. We have also slightly modified our manuscript to clarify our motivations.
>
> ----
>
> > **[Q2]** ***The technical depth seems weak. Table 1 is a good summary, but it also shows the weakness of the proposed model.***
>
> **[Response]** Our paper is mainly focused on the study of a new paradigm where the model capacity of federated learning is broken through by adopting a larger server model via selective knowledge fusion. This has never been proposed before and we demonstrate with comprehensive experiments that our framework greatly surpass (by over 20\%) existing frameworks' performance on both server and clients side. Therefore we think our contributions are significant. Our approaches have certain limitations, including that it depends on a labeled public dataset, but is still effective and useful for other scenarios where some labeled public dataset is available.

---

> > ### Comment · Reviewer_tN4o · 2021-11-29
> > **thanks authors response**
> >
> > I have read the author's response and the comments from other reviewers. I focused on the dialogue between the authors and reviewer Cj8x but concerns still remain (w.r.t. weakness points 1-6).

---

> > > ### Author Response · Authors · 2021-11-30
> > > **Thanks Letter for Review**
> > >
> > > Dear Reviewer,
> > >
> > > Thanks for your response and your valuable time. We have further addressed reviewer Cj8x's concerns on weakness points 1-6 (See our responses below) and the corresponding changes are made to our manuscript based on reviewer Cj8x's feedback. If you have other specific concerns about our work in addition to other reviewers' comments, we would really appreciate that **if you can point them out case-by-case and provide more in-depth suggestions**, like other reviewers, since those suggestions have helped us to make our work better.
> > >
> > > Best Regards.

---

### Decision · Program_Chairs · 2022-01-20

**Decision:**

Reject

**Comment:**

This paper proposes a knowledge distillation strategy to enable the use of a large server-side model in federated learning while satisfying the computation constraints of resource-limited clients. The problem is relevant and well-motivated, and the paper presents compelling experimental results to support the proposed strategy. However, reviewers had the following major comments suggestions/:
1) The theoretical analysis section needs improvement in terms of the technical depth and rigor
2) Better explanation of how the proposed strategy compares with previous works/baselines
3) Considering the privacy and scalability properties of the proposed strategy.

The paper generated lots of constructive post-rebuttal discussions between the authors and the reviewers, and I believe the authors received several ideas to improve the work and appreciated the reviews. One of the reviewers increased their score. However, based on the current scores, I still recommend rejection. I do think the paper has promise, and with improvements, the revised version will make an excellent contribution.